# Wintertime subarctic new particle formation from Kola Peninsula sulphur emissions

Mikko Sipilä[1], Nina Sarnela[1], Kimmo Neitola[1], Totti Laitinen[1], Deniz Kemppainen[1], Lisa Beck[1], Ella-Maria Duplissy[1], Salla Kuittinen[1,2], Tuuli Lehmusjärvi[1], Janne Lampilahti[1], Veli-Matti Kerminen[1], Katrianne Lehtipalo[1,2], Pasi P. Aalto[1], Petri Keronen[1], Erkki Siivola[1], Pekka A. Rantala[1], Douglas R. Worsnop[1,3], Markku Kulmala[1], Tuija Jokinen[1], Tuukka Petäjä[1]

[1]Institute for Atmospheric and Earth System Research, PO. Box 64, 00014, University of Helsinki, Finland
[2]Finnish Meteorological Institute, 00560, Helsinki, Finland
[3]Aerodyne Research Inc. 01821 Billerica, MA, USA

*Correspondence to*: Mikko Sipilä (mikko.sipila@helsinki.fi)

**Abstract.** Metallurgical industry in Kola Peninsula, North-West Russia, form, after Norilsk, Siberia, the second largest source of air pollution in the Arctic and sub-Arctic domain. Sulphur dioxide ($SO_2$) emissions from the ore smelters are transported to wide areas, including Finnish Lapland. We performed investigations on concentrations of $SO_2$ and aerosol precursor vapours, aerosol and ion cluster size distributions together with chemical composition measurements of freshly formed clusters at the SMEAR I station in Finnish Lapland relatively close (∼300 km) to the Kola Peninsula industrial sites during the winter 2019-2020. We show that highly concentrated $SO_2$ from smelter emissions is converted to sulphuric acid ($H_2SO_4$) with sufficient concentrations to drive new particle formation hundreds of kilometres downwind from the emission sources, even at very low solar radiation intensities. Observed new particle formation is primarily initiated by $H_2SO_4$–ammonia (negative-) ion-induced nucleation. Particle growth to cloud condensation nuclei (CCN) sizes was concluded to result from sulphuric acid condensation. However, air mass advection had a large role in modifying aerosol size distributions, and other growth mechanisms and condensation of other compounds cannot be fully excluded. Our results demonstrate the dominance of $SO_2$ emissions in controlling winter-time aerosol and CCN concentrations in the subarctic region with a heavily polluting industry.

## 1 Introduction

Sulphur dioxide ($SO_2$) is one of the main air pollutants, influencing the acidification of soils and freshwaters, defoliation and reduced vitality of forests, atmospheric aerosol formation, cloud properties, and adverse health effects by air pollutants. Anthropogenic $SO_2$ originates primarily from the combustion of fossil fuels at power plants, other manufacturing complexes

and ships, as well as from the smelting of sulphur-containing mineral ores. Because of the severe environmental and health effects by $SO_2$, efforts have been made in order to suppress its emissions to the atmosphere. While the global $SO_2$ emissions have not shown any rapid decay, emissions in e.g. OECD (Organisation for Economic Co-operation and Development) countries have decreased significantly within last 3 decades (Solarin and Tiwari, 2020).

The metallurgical industry with large-scale smelter complexes in the Kola Peninsula, North-West Russia, form the second largest source of air pollution in the Arctic and sub-Arctic region. Smelters emit large quantities of $SO_2$, metals and particulate matter to the atmosphere. These pollutants, especially $SO_2$, have large impacts on both atmosphere and biosphere in the surrounding area, including the eastern parts of Finnish and Norwegian Lapland. In the close proximity of industrial plants, these pollutants have literally destroyed ecosystems, creating "industrial deserts" (Paatero et al., 2008). Though emissions have significantly decreased from about 600 kilotons $yr^{-1}$ in early 1990's (Tuovinen et al., 1993; Ekimov et al., 2001), partly because of the collapse of Soviet Union and related socio-economical changes in Russia, they are still high. Today, the $SO_2$ emissions from Kola Peninsula are around 200 kilotons $yr^{-1}$ (Barentz Observer, 2019), far higher than the $SO_2$ emissions of the whole Finland (37 kilotons $yr^{-1}$ in 2017). Though vast, Kola emissions are still far behind the emissions of the World's number one $SO_2$ polluter, Norilsk (Krasnoyarsk Krai, Northern Siberia), with enormous 1.5 megatons $yr^{-1}$ emission rates (Barentz Observer, 2019). Together with the few other smaller-scale industrial complexes, these smelters are almost the sole local sources of air pollution in the very sparsely-populated (sub-)Arctic Eurasia, and therefore understanding their role in atmospheric chemistry and physics is of great importance.

Most of the atmospheric sulfate is formed from $SO_2$ in a liquid phase in cloud droplets, and these droplets either evaporate leading to sulfate aerosol production or precipitate as acid rain. However, with very high concentrations of $SO_2$ downwind the Kola Peninsula area, high production rates of gas-phase sulphuric acid ($H_2SO_4$) due to photochemical oxidation of $SO_2$ is expected. The $H_2SO_4$ vapour can, in turn, contribute to atmospheric new particle formation (NPF) via nucleation and subsequent particle growth even up to sizes of cloud condensation nuclei (CCN) by further condensation of $H_2SO_4$ and potentially some other vapours (e.g. Weber et al., 1995; Kirkby et al., 2011; Jokinen et al., 2018). Atmospheric NPF is an important process because, according to model simulations, it accounts for more than a half of atmospheric CCN formation globally (Merikanto et al., 2009; Gordon et al., 2017). At high latitudes, the contribution of NPF has been estimated to be even larger, reaching >90% of the cloud level CCN in the high Arctic and approximately 70-80% in our study area, the sub-Arctic zone of Northern Finland and North-Western Russia (Gordon et al., 2017).

Vehkamäki et al. (2004) were the first to report observations of NPF (> 8-nm diameter particles) at the Värriö SMEAR I field station in eastern Lapland, Finland, relatively close to the Kola Peninsula smelter complexes. Their results on the

contribution of $SO_2$ pollution were not completely definitive, so that during the four years of measurements 15 out of the 147 observed NPF events were concluded to be explained by $SO_2$ pollution plumes. Kyrö et al. (2014) recorded particle number size distributions down to 3 nm in diameter and showed that NPF is connected to high concentrations of $SO_2$. They observed NPF even during winter in almost dark conditions, indicating that during episodes of very high concentrations of $SO_2$, a sufficient fraction of it is converted to $H_2SO_4$ in the gas phase even in very low solar radiation levels to initiate NPF. However, to date, no reports on quantification of sulphuric acid concentrations by direct measurements, nor detailed mechanisms and chemical compounds involved in NPF, in this area exist.

While observation of atmospheric NPF has been reported in hundreds of publications since the times of John Aitken (Aitken, 1900), the details, i.e. the dynamics and contributing compounds, of NPF have been experimentally resolved only in a limited number of atmospheric environments. Pioneering studies include the observations by Weber et al. (1995) on the connection between sulphuric acid and atmospheric nucleation, and the first report on ion-induced nucleation and simultaneous detection of sulphuric acid anion clusters using a mass spectrometer by Eisele et al. (2006). Parallel to the field work, several laboratory investigations by the same research groups probed the properties of sulphuric acid – water and sulphuric acid – ammonia – water clusters and their potential role in new particle formation (Ball et al., 1999; Hanson and Eisele, 2002).

Later advances in understanding the *molecular* steps of nucleation and growth in the atmosphere include the discovery that iodic acid ($HIO_3$) is primarily responsible for nucleation and growth in coastal areas and in the vicinity of the Arctic sea ice (Sipilä et al., 2016; Baccarini et al., 2020). Jokinen et al. (2018) demonstrated that in coastal Antarctica, $H_2SO_4$ originating from the oxidation of dimethyl sulphide (DMS, emitted by pelagic phytoplankton) and ammonia ($NH_3$, from penguin colonies) nucleate via a negative ion-induced mechanism, with sulfuric acid condensation accounting for most of the subsequent particle growth. Further observations on nucleation mechanisms indicate the key role of highly oxidized organic molecules, HOMs (Ehn et al., 2014), in NPF during the spring-summer time in a boreal forest environment (e.g. Kulmala et al., 2013; Rose et al., 2018) and in the mid-latitude continental free troposphere (Bianchi et al., 2016) in parallel with sulfuric acid – ammonia nucleation (Bianchi et al., 2016; Schobesberger et al., 2015; Yan et al., 2018). Amines, especially dimethyl amine, were found to contribute to the initiation of nucleation in polluted urban air (Yao et al., 2018; Brean et al., 2021; Cai et al., 2021). In addition to these, yet rare molecular-level atmospheric observations, several laboratory studies have investigated the details of these nucleation mechanisms (e.g. Kirkby et al., 2011; Almeida et al., 2013; Kürten et al., 2014; Kirkby et al., 2016). Recent laboratory studies that probed nucleation of nitric acid and ammonia suggest that this mechanism may contribute to new particle formation and growth, especially in the upper troposphere (Wang et al., 2020).

Mass spectrometers (Junninen et al., 2010; Jokinen et al., 2012) and air ion spectrometers (Mirme and Mirme, 2013), have largely facilitated the recent progress in the field of atmospheric NPF. By utilizing them in conjunction with aerosol and meteorological observations, this work aims to shed light on the molecular steps of NPF resulting from (sub-)Arctic air pollution during wintertime. Investigations were carried out at the SMEAR I research station in Värriö strict nature reserve in Finnish Lapland close to the industrial plants (most of them located approximately 300 km east from the station) of Kola Peninsula, north-west Russia, during the winter 2019–2020.

## 2 Methods

### 2.1 Site and time of the study

Measurements were carried out at the Värriö SMEAR I research station (Hari et al., 1994) located in Värriö strict nature reserve, Finnish Lapland, in the vicinity (5 km) of the Russian border (Fig. 1) (67°45′19″N 29°36′37″E). The station stands on a top of a hill (390 m a.s.l.), being surrounded by untouched pine and spruce forests, bogs, fells, small lakes and rivers. Several large smelter complexes are located ~300 km away from to the station on the Russian side of the border, while on the Finnish side no smelters or other large-scale energy intensive (polluting) industrial plants are located within a distance of several hundreds of km. The closest, relatively small coal burning plant is located 550 km away. The SMEAR I station was set up in 1991 for monitoring air pollution, especially sulphur dioxide ($SO_2$) originating from the Kola Peninsula smelters. In this work we present 4.5 months of data from winter time, covering the period 1 November 2019 – 16 March 2020.

### 2.2 Instrumentation

The aerosol number size distribution between 3 and 750 nm of particle diameter was recorded by a twin differential mobility particle sizer (DMPS) (Aalto et al., 1999), comprising Hauke-type differential mobility analyzers (lenghts 110 and 280 mm) and TSI-3776 and TSI-3772 condensation particle counters (TSI Inc., Shoreview, MN, USA) as detectors. The DMPS measuring 3–10 nm particles malfunctioned during 9–10 and 14–27 January, resulting in the loss of data from this size range on those days.

The number size distribution of charged particles and molecular clusters between 0.8 and 40 nm was recorded by a Neutral Cluster and Air Ion Spectrometer (NAIS, Airel Ltd., Estonia; Mirme and Mirme, 2013).

Aerosol precursor vapour concentrations of $H_2SO_4$, methane sulphonic acid (MSA), $HIO_3$ and HOM were measured by a nitrate ion – Chemical Ionization -Atmospheric Pressure interface – Time-Of-Flight mass spectrometer (CI-APi-TOF, Jokinen et al., 2012), equipped from 26 January onwards with a switcher inlet, with which the instrument can switch between

chemical ionization (CI) operation mode and natural ion detection mode. This instrument was calibrated in the CI-mode for sulphuric acid, as described by Kürten et al. (2012). The same calibration coefficient was used for the reported MSA and $HIO_3$ concentrations. However, the instrument was not fully operational at all the times during the measurement campaign, so a significant fraction of data (including all data collected before 25 December and a long period in January) was disregarded.

$SO_2$ was recorded with a TEI 43 i-TLE pulsed fluorescence analyzer, $O_3$ by a TEI 49 i photometric analyzer, and $NO_x$ by a TEI 42C TL chemiluminescence analyzer with photolytic $NO_2$-to-NO converter, all manufactured by Thermo Fischer Scientific (Franklin, MA, USA). The wind speed and direction as well as the air temperature were measured with a Vaisala WTX sensor 16 m above ground level.

## 2.3 Nucleation rate calculation

Negative (-) and positive (+) ion-induced nucleation rates of 1.5 nm particles, $J^{-/+}_{1.5}$, were calculated assuming a steady-state between formation and loss of particles in the size range of 1.5 and 2.5 nm:

$$J^{-/+}_{1.5} = \frac{dN^{-/+}_{1.5-2.5}}{dt} + \frac{GR_2}{\Delta d_p}N^{-/+}_{1.5-2.5} + CoagS\, N^{-/+}_{1.5-2.5} + k_{rec}N^{+/-}_{<1.5}N^{-/+}_{1.5-2.5} \qquad , \qquad (1)$$

where $N^{-/+}_{1.5-2.5}$ is the total concentration of negative or positive ions in the size range between 1.5 and 2.5 nm, filtered using Matlab's Savitzky-Golay second-order filtering method to remove the instrument noise, $k_{rec}$ is the recombination coefficient between negative and positive small ions which was here approximated by a size-independent constant of $1.6 \times 10^{-6}$ cm$^3$ s$^{-1}$ (Tammet, 1995), $N^{+/-}_{<1.5}$ is the concentration of positive or negative sub-1.5 nm cluster ions. $GR_2$ is the 2 nm particle growth rate, $\Delta d_p$ is the width of the size interval for which the concentration is defined ($\Delta d_p$ = 2.5 nm – 1.5 nm = 1 nm), and $CoagS$ is the coagulation sink of 2 nm particles to the pre-existing background aerosol population. CoagS was calculated from the following equation:

$$CoagS = \sum_{i=1}^{n} K_{2nm,\,i}N_i \qquad (2)$$

where $N_i$ is the concentration of particles in the channel $i$ of DMPS and $K_{2nm,i}$ is the coagulation coefficients between a 2-nm particle and a particle in the size bin $i$ calculated based on Seinfeld and Pandis (1998).

An accurate determination of the particle growth rate for 2 nm particles from the size distribution is challenging, and therefore $GR_2$ was approximated by assuming irreversible sulphuric acid condensation as the sole mechanism of growth similar to Jokinen et al. (2018) and Beck et al. (2021), and it was calculated according to the formula given by Stolzenburg et al., (2020):

$$GR_2 = 1.45 \cdot \left(2.68 \cdot \left(\frac{d_p}{nm}\right)^{-1.27} + 0.81\right) \bullet \left[H_2SO_4\right] \bullet 10^{-7} molec.^{-1} cm^3 \qquad . \qquad (3)$$

Here the pre-factor 1.45 accounts for dipole - charge interaction in charged particle growth (Stolzenburg et al., 2020). The justification for this approach will be discussed later. A more standard method for the GR determination is to approximate the $GR_2$ by the average growth rate of the formed particle population, including mainly particles grown far above the 2 nm size, during a few hours starting from the beginning of the event as demonstrated in Figure 2A. This method leads to the average GR of 4.5 nm h$^{-1}$. However, this approach neglects the effect of air mass advection which, as will be discussed later, may largely determine the time development of the size distribution and thus also the apparent growth.

Rather than the average GR of the whole particle population, the 50-% appearance time method (Lehtipalo et al. 2014) could be used to estimate the growth rate of nucleating clusters in the size range of 1.3–2.7 nm (Figure 2B). Here, the cluster appearance time in each size channel represents the time when cluster concentration reaches 50% of its maximum concentration during the event. The growth rate can be assessed from the cluster diameter vs. appearance time curve (black line), resulting in ~0.35 nm h$^{-1}$ during 10:10-11:30 and ~1.8 nm h$^{-1}$ during 11:30-11:50, and in the average growth rate of ~0.9 nm h$^{-1}$ during the period 10:10-11:50. A drawback of this analysis is that the GR cannot be obtained for the period where the concentration has passed the 50%-threshold or the period of decaying concentration. Furthermore, the temporal variability of GR cannot be properly obtained due to fluctuations in the data. Nevertheless, the above-mentioned values of 0.35–1.8 nm h$^{-1}$ can be compared to those obtained from Eq. 3, which yields the maximum GR of 0.51 nm h$^{-1}$ around the noon on the example day (29 January 2020). A comparison to the value of 1.8 nm h$^{-1}$ obtained from the 50%-appearance time method slightly before noon leads to a factor of ~3.5 difference in $GR_2$ on that day, which, in turn, is reflected in a 22% difference in the calculated nucleation rate (Eq. 1). The effect of the different approaches to determine GF is visualized in Figure S1. To conclude, ion-induced nucleation rate calculation is not very sensitive to $GR_2$ because ion-ion recombination term (Eq. 1) dominates the loss in our conditions.

## 2.4 Sulphuric acid proxy calculation

Because of significant gaps in the measured data, $[H_2SO_4]$ was also calculated using a proxy developed by Dada et al. (2020). This proxy takes into account the oxidation of $SO_2$ to $H_2SO_4$ both by OH (estimated from the global radiation intensity) and stabilized Criegee Intermediates (estimated from monoterpene and ozone concentrations; Sipilä et al., 2014), as well as losses of $H_2SO_4$ by dimerization (negligible in observed concentrations) and condensation onto pre-existing aerosol particles (the primary loss term). Unfortunately, there are no VOC measurements available at SMEAR I, but, because the data were collected during winter well outside of the growth period, we assumed the monoterpene concentration to be zero. Global radiation measurements showed unexplained fluctuations (maybe caused by low solar zenith angles or freezing of the sensor) during the measurement period, and therefore we used UVB radiation and the relation between UVB and global radiation determined by Dada et al. (2020). During the times when CI-APi-TOF was operational, the agreement between the measured and calculated concentrations was good (Figure S2) with mean concentrations agreeing within 8%. The obtained correlation coefficient was R = 0.790 and coefficient of determination $R^2 = 0.624$.

## 2.5 Trajectory analyses

Trajectories were calculated by using the Hybrid Single-Particle Lagrangian Integrated Trajectory model HYSPLIT (Stein et al., 2015) with GFS 0.25 degree meteorology as an input. We calculated 24-hour backward trajectories arriving at 50 and 250 meters above ground level for the period 28 January 2020 at 00:00 UTC to 30 January 2020 at 00:00 UTC, arriving every 6 hours. The trajectory calculations included mixing layer depth along the trajectory.

## 3 Results and Discussion

### 3.1 New particle formation during the measurement period.

Figures 3a and 4a depict the aerosol number size distribution between 3 and 700 nm, as recorded by the DMPS. Several new particle formation events were observed during the measurement period. Clear NPF events with the concentration of 4–10 nm particles, $N_{4-10nm}$, (Figures 3b & 4b) exceeding 50 $cm^{-3}$ are marked with grey shadings. Since the DMPS data on sub-10 nm particles were missing from 9–10 band 14–27 January, $N_{4-10nm}$ could not be derived for those time periods. Still, at least on 18 and 19 January we can see that NPF eventually produced particles larger than 10 nm in diameter. The observed NPF

events coincided mostly (ca. 50% of the cases) with clearly easterly (~90°) winds (Figures 3c and 4c), and with elevated $H_2SO_4$ concentrations (Figures 3d and 4d) calculated based on Dada et al. (2020). $H_2SO_4$ concentrations depend, besides condensation sink and UVB radiation, on the $SO_2$ concentration that is connected to both wind direction and air mass origin. Clear examples of such, $SO_2$ pollution-driven NPF events are, for example, those occurring during three consecutive days on 10–12 November, 2019, on 28 and 29 January, 2020, and on 13 March, 2020. The data from the period 28–29 January are discussed in detail below, whereas the data from the other two exemplary periods are presented in the Supplement (Figures S3–S11).

Not all the NPF events occurred during the easterly winds. The events observed close to the mid-winter, from early December until early January, occurred with westerly winds or during the transition of the wind direction from west to east, in relatively low concentrations of $SO_2$ and in the virtual absence of daylight and $H_2SO_4$. These low-$H_2SO_4$ mid-winter events were observed to start from sizes larger than a few nm, which means that nucleation did not take place *in situ* in the surroundings of the SMEAR I station. Those particles had thus formed elsewhere, and were transported to the measurement site by either horizontal advection or vertical downdrafts from above the mixed layer. Compared with particles of a few nm in diameter, gas-phase $H_2SO_4$ is lost much more rapidly onto pre-existing particles after its production ceases, so the lack of $H_2SO_4$ is not excluding its primary role in NPF, even though it is not supporting such role either. Some of the mid-winter NPF events coincided with elevated $SO_2$ concentrations, suggesting that sulphuric acid may have be formed in the measured air mass earlier. Some other mid-winter NPF events, especially the relatively strong NPF event on 3 December presented in the Supplement (Figures S9-S11), occurred in a virtual absence of $SO_2$, suggesting that sulphuric acid had not been formed to a significant extent in that air mass. Currently, we thus cannot explain the mechanism of NPF on that day. However, the $NO_2$ concentration was slightly (Figure S9) elevated in the measured air mass, which might have been connected with the elevated source of particles. Nevertheless, most of the NPF events seemed to be connected with the presence of $H_2SO_4$.

## 3.2 Case study 28th – 29th January 2020

To resolve the details of new particle formation and growth, we focus on 3 time periods with clear signs of nucleation and particle growth. Here we show results from analysis of a 2-day period of 28–29 January 2020. To demonstrate that this 2-day period is not only a unique observation, we represent data from the time period 10–12 November, 2019 (Figs. S3–S5) and from 13 March, 2020 (Figs. S6–S8) in the Supplement. The data from the event on 3 December, 2019, which differs from the overall picture, are also depicted in the Supplement (Figs. S9–S11).

### 3.2.1 Meteorological situation and trace gas concentrations

Throughout the period of 28–29 January, the wind was blowing from the East (~50–150°) (Fig. 5a). The ambient temperature ranged from –16°C to –28°C (Figure 5b). The sky was clear but, because of the low solar zenith angle (maximum 4.4° at the noon of 29 January), the UVB radiation intensity needed for a photochemical $H_2SO_4$ formation reached only 35 mW m$^{-2}$ (summertime UVB radiation intensity maxima at Värriö are >2000 mW m$^{-2}$). The HYSPLIT back trajectory calculations showed that the air masses arriving between 28 January at 6:00 and 30 January at 00:00 passed the industrial areas of the Montchegorsk region (Fig. 6).

At around 3:00 on 28 January, coinciding with the change in the air mass origin to the Montchegorsk–Kandalaksha region (Fig. 6) a few hours after the change in the wind direction from west to east in the evening of 27 January, air pollutant levels started to increase steeply (Fig. 5d). During the course of the day, both $SO_2$ and $NO_2$ concentrations increased ca. 2 orders of magnitude, with [$SO_2$] peaking at 27 ppb and [$NO_2$] peaking at 7 ppb. The ozone ($O_3$) concentration declined from about 40 ppb to a 30 ppb range. To put the high level of $SO_2$ concentrations into some perspective, the highest concentration recorded in the Helsinki Metropolitan area was 8.4 ppb (24 µg m$^{-3}$, 1-hour average) in 2019 and the yearly-average concentration was about 0.2 ppb (~0.5 µg m$^{-3}$) (Helsinki Region Environmental Services Authority, 2020). The yearly-average $SO_2$ concentration at the SMEAR I station in 2019 was 1.1 ppb.

### 3.2.2 Aerosol precursors

Despite the low UVB radiation, required for $O_3$ photolysis that initiates the $H_2SO_4$ production via OH radical formation, the $H_2SO_4$ concentration increased from close to the lowest detection limit values of ~10$^5$ molecules cm$^{-3}$ up to 8×10$^5$ molecules cm$^{-3}$ during 29 January and up to 1.5·×10$^6$ molecules cm$^{-3}$ on 29 January (Fig. 7e). Because the OH production rate must have been low, a high $SO_2$ concentration is a perquisite for the $H_2SO_4$ production during cold and dark winter months. While stabilized Criegee Intermediates (sCI) formed in alkene ozonolysis can oxidize $SO_2$ to produce $H_2SO_4$ during summertime (Mauldin et al., 2012; Sipilä et al., 2014), alkene (terpene) emissions from the vegetation and thus the sCI production rate are negligible during the winter season. The proxy calculations agree well with the measured sulphuric acid concentration on 29 January but show clearly higher values on 28 January. The cause of the disagreement on 28 January is probably the stable and shallow boundary layer. The temperature gradient close to the surface was almost +0.2 °C m$^{-1}$ at noon on 28 January (Fig. 5b). Solar radiation from close to the horizon does not penetrate efficiently inside the canopy, so the UVB measured above the canopy and used in the proxy calculation does not reflect the situation at the ground level. Sulphuric acid produced above the canopy, on the other hand, does not mix downwards due to the strong temperature inversion and calm winds. On

29 January, the temperature gradient was absent or slightly negative, allowing the surface air to mix with the air above canopy.

Besides $H_2SO_4$, also minute signals of iodic acid ($HIO_3$) were observed during the day (Fig. 7e). The exact production mechanism of $HIO_3$ remains globally unknown despite the emerging evidence on its critical role in new particle formation especially in the Arctic regions (Sipilä et al., 2016; Baccarini et al., 2020). Methane sulphonic acid (MSA), that has been observed in larger aerosol particles (Beck et al., 2021) and that could potentially also contribute to NPF, hardly exceeds the detection threshold. This was expected since MSA originates from dimethyl sulphide (DMS) photo-oxidation. DMS ends up into the air mainly from the metabolism of pelagic phytoplankton during summer months, not during the dark winter. No other condensable vapours, such as HOM which dominate the new particle growth in the summer-time boreal forest environment (Ehn et al., 2014), were observed during this case study period or during other periods depicted in Supplement.

### 3.2.3 New particle formation

*Ion size distribution*

Figures 7a and 7b show the NPF events on 28 and 29 January, as observed by the NAIS operated in the ion mode. Omnipresent small, < 1.5 nm ions are continuously produced by the galactic cosmic radiation, terrestrial gamma radiation and gas phase radon decay. Approximately at 11:00 on 28 January, coinciding with the increase of the $H_2SO_4$ concentration, small negative cluster ions started to grow, which is seen as small increases in the ~1.5–2 nm negative ion concentration. During their growth beyond ~2 nm in diameter, those clusters were neutralized by collisions with positively charged ions, so that they disappeared from the spectrum. They still obviously continued to grow in size, as charged particles reappeared in the spectrum after reaching some 5 nm in diameter when diffusion charging becomes effective enough; an equilibrium charging state for 2 nm particles is 0.8%, while 5 nm particles are charged with an efficiency of 2.3% and out of 20 nm particles 11% are negatively charged (Wiedensohler et al., 2012). Opposite to the negative ions, positive cluster ions did not grow in size. Larger, > 5nm positive particles (charged by diffusion charging during the course of their growth) grew similarly to the negative ones. On 29 January, with clearly higher $H_2SO_4$ concentrations, the appearance of >1.5 nm negative clusters was more pronounced, suggesting higher nucleation rates and critical role $H_2SO_4$ in the initial steps of NPF. Positive cluster ions were again only bystanders and did not contribute to nucleation. This observation suggests that negative ion-induced nucleation is the primary pathway to NPF similar to $H_2SO_4 - NH_3 (- H_2O)$ ion-induced nucleation observed by Jokinen et al. (2018) in Antarctica and Kirkby et al. (2011) in CERN CLOUD chamber experiments. However, due to lack of information on neutral ~1.5–3 nm cluster concentrations, this observation alone does not exclude parallel neutral nucleation mechanisms.

*Nucleation rates*

Even though a weak growth of the small negative ions at around noon on 28 January is visually observable in Fig. 7a, the concentration of clusters in the 1.5–2.5 nm size range ($N^-_{1.5-2.5}$) is hardly distinguishable from the noise (Fig. 7c). The nucleation rate, calculated using the filtered concentration data, only slightly exceeded the baseline (caused by presence of minute, almost omnipresent signal from ion clusters extending above 1.5 nm but which is not connected to sulphuric acid nucleation), being approximately 0.005 cm$^{-3}$ s$^{-1}$ with a high relative uncertainty (Fig. 7d). On 29 January, with a 2.3-fold sulphuric acid concentration, the concentration of 1.5–2.5 nm negative clusters was well above the instrument noise, reaching 20 cm$^{-3}$ around the noon. The nucleation rate peaked at 0.067 cm$^{-3}$ s$^{-1}$. The ambient temperatures during nucleation (~ noon) were close identical, around –22°C, in both days and therefore they can be directly compared. An approximately 10-fold difference in the nucleation rate between the two days accompanied by a factor of 2.3 difference in the sulphuric acid concentration is in line with the results from the CLOUD-chamber experiment on sulphuric acid – ammonia – water nucleation (Kirkby et al., 2011). The so-called "slope" that approximately (not exactly in real atmospheric situations) equals to the number of sulphuric acid molecules in the critical cluster (Vehkamäki et al., 2012) is given as:

$$Slope = \frac{dlogJ^-_{1.5}}{dlog[H_2SO_4]}$$

and yields a value of 2.9 for the numbers discussed above. Though this value is subject to a significant uncertainty, it would agree with observations of Kirkby et al. (2011) and parameterizations by Dunne et al. (2016) which yield a "slope" of approximately 3 under similar conditions as visualized in Fig. 8. In the same figure, data from all the days with clearly observable ion-induced nucleation are depicted. There, hourly average nucleation rates $J_{1.5}^-$, that exceed a threshold value of $J_{1.5}^- = 0.01$ cm$^{-3}$ s$^{-1}$, are plotted against the concurrent calculated sulphuric acid concentration and air temperature. Calculated nucleation rates, $J_{gcr}$, represent the total nucleation rates (ion-induced plus neutral) at different temperatures and ammonia concentrations under the influence of galactic cosmic radiation (GCR) producing ions with the fixed rated of 1.8 ion pairs cm$^{-3}$ s$^{-1}$. Negative ion-induced nucleation, however, should be the dominant mechanism under these conditions (Kirkby et al., 2011), so these results can be compared. Our data are reasonably close to the range predicted by the parameterization, considering that this simple calculation does not include air mass transportation, vertical mixing, terrestrial radiation sources, or any other real-world phenomena. Also, sources and concentration of ammonia in our study area are unknown.

*Cluster time series*

To confirm the role of sulphuric acid and to solve the contribution of ammonia to the nucleation process, we measured the negative ion cluster composition and signal intensity with the APi-TOF in the ion mode without chemical ionization. The time series of the most abundant clusters show the appearance of $HSO_4^-$ -ion in the morning of 28 January, together with an increasing $[H_2SO_4]$ accompanied with a decay of $NO_3^-$ -ion signal which typically dominates the anion spectrum at low $[H_2SO_4]$ and low $[HIO_3]$ globally (Fig. 7f). Since $H_2SO_4$ is a stronger acid than $HNO_3$, the proton transfer from $H_2SO_4$ to $NO_3^-$ explains the observed behaviour when $[H_2SO_4]$ started to rise. When $[H_2SO_4]$ still increased during the course of the day, $(NH_3)_m(H_2SO_4)_nHSO_4^-$ -clusters started to form. The cluster signals peaked at around the noon coinciding with the highest $[H_2SO_4]$ and $N^-_{1.5-2.5}$, after which they started to decay. On 29 January, the same behaviour was observed, but with somewhat stronger cluster signals due to the higher $[H_2SO_4]$.

*Cluster composition*

To get more insight into the chemical composition of clusters, the ion-cluster mass spectrum was integrated over 4 hours (2 hours effective data collection due to switching between CI and ion-inlet). The resulting spectrum is presented in Fig. 9 by means of a mass defect plot, where the mass-to-charge ratio (*m/z*, unit Th) corresponds – with only singly-charged ion clusters present in the air – to the mass of the cluster (*m*, unit Da, equal to unified atomic mass unit, u). Mass defect is the mass difference (in Th or Da) between the exact mass of a cluster and the integer mass defined as the sum of nucleons in the atomic nuclei of this cluster. For example, the exact mass of a $HSO_4^-$ -ion that has 97 nucleons is 96.960103 Da and the mass defect is thus 0.039896 Da. The area of a dot is proportional to the logarithm of the observed signal intensity. In the mass defect plot, each addition of a molecule or atom is represented by a vector. An addition of e.g. $H_2SO_4$, with a negative mass defect, leads to an increasing mass and a decreasing total mass defect, while an addition of a positive mass defect $NH_3$ molecule leads to an increasing total mass defect. Successive additions of certain molecules to an ion results in a straight line in the mass defect plot, so that different cluster formation pathways are readily distinguishable from that plot.

In Fig. 9, the largest signals are associated with the omnipresent nitrate ion and its cluster with nitric acid ($NO_3^-$ and $HNO_3 \cdot NO_3^-$). The rest of the small (<180 Da) ions are mainly different sulphur species, with bisulphate ion partly clustered with nitric acid ($HSO_4^-$ and $HNO_3 \cdot HSO_4^-$) being the most abundant ones. Other small sulphur ions present in the spectrum are $SO_4^-$, $SO_5^-$, $HNO_3 \cdot SO_3^-$ and $HNO_3 \cdot SO_4^-$. Deprotonated iodic acid ($IO_3^-$) and its nitric acid cluster, ($HNO_3 \cdot IO_3^-$) are also abundant. Despite the presence of multiple different types of these core ions, their initial growth is solely caused by the

attachment of sulphuric acid molecules. We observed clusters with 1–4 $H_2SO_4$ molecules attached to the $SO_4^-$ - ion, one $H_2SO_4$ molecule attached to the to $SO_5^-$ and $SO_3^-$ - ions, and 1–3 $H_2SO_4$ molecules attached to the $IO_3^-$ -ion. For simplicity, we assume that the negative charge remains in the core ion. This is not necessarily true, but $H_2SO_4$ may lose a proton e.g. to $IO_3^-$ , resulting in the composition of $HIO_3 \cdot (H_2SO_4)_{n-1} \cdot HSO_4^-$ instead of $^-(H_2SO_4)_n \cdot IO_3^-$. Furthermore, water, if present in the clusters, efficiently evaporates in the vacuum of a mass spectrometer and therefore information on the role of water in the cluster formation is lost.

None of the clusters discussed above seem to adopt ammonia efficiently enough for their signals to exceed the detection threshold of the APi-TOF (mass dependent, $\sim 10^{-3}$ to few $10^{-3}$ ions/second for 2 hour integration for m/z > 400 Th). Only clusters made solely of sulphuric acid with a bisulphate ion ($HSO_4^-$) as a core seem to efficiently attach ammonia, resulting in the formation of $(NH_3)_m \cdot (H_2SO_4)_n \cdot HSO_4^-$ -clusters (n>=3). This sequential addition of $NH_3$ and $H_2SO_4$ has been shown to be an effective (ion-induced) cluster formation and growth mechanism in coastal Antarctica (Jokinen et al., 2018) as well as a secondary pathway in the free troposphere (Bianchi et al., 2016) and in the spring/summer time southern Finland boreal forest (Yan et al., 2018).

Our results on negative cluster composition can be compared to the results from the CLOUD experiment at –25°C for varying $NH_3/H_2SO_4$ ratios (Schobesberger et al., 2015). Based on that data, with the $NH_3/H_2SO_4$ ratio exceeding approximately 100, both cluster composition and nucleation rate saturate (Kirkby et al., 2011), and become unaffected by further increases of the $NH_3$ concentration. In those conditions, a cluster comprising 3 molecules of sulphuric acid on a bisulphate ion, $(NH_3)_n \cdot (H_2SO_4)_3 \cdot HSO_4^-$, contains on average approximately $n \sim 1$ molecules of ammonia, whereas a cluster composed of 4 molecules of sulphuric acid and a bisulphate ion, $(NH_3)_n \cdot (H_2SO_4)_4 \cdot HSO_4^-$, carries on average approximately $n \sim 1.5$ $NH_3$ molecules (Schobesberger et al., 2015). In our case (Fig. 9), corresponding average ammonia numbers were $n \sim 0.4$ and $n \sim 0.8$ for $(NH_3)_n \cdot (H_2SO_4)_3 \cdot HSO_4^-$ and $(NH_3)_n \cdot (H_2SO_4)_4 \cdot HSO_4^-$, respectively, which would suggest that the $NH_3/H_2SO_4$ ratio in our case was well below 100, and likely below 10 (Schobesberger et al., 2015). If true, that would indicate an ammonia concentration of the order of $\sim 10^7$ molecules $cm^{-3}$, or $\sim 1$ pptv. However, cluster fragmentation inside the mass spectrometer can be totally different between our experiment and Schobesberger et al. (2015) study, and therefore not any conclusions on ammonia concentrations should be drawn. Nevertheless, if the $NH_3/H_2SO_4$ ratio were low, the system would not saturated with respect to $NH_3$ and the nucleation rate should therefore be sensitive to both $H_2SO_4$ *and* $NH_3$ similar to Jokinen et al. (2018). This, together with unknown effects of cluster fragmentation, highlight the importance of understanding $NH_3$ sources, transportation and atmospheric mixing ratios down to ppt levels for a proper description of new particle formation also in the subarctic region. Unfortunately, $NH_3$ concentrations in the range of 1 pptv are not (reliably) detectable with any present-day measurement technology.

The present analysis shows that the sulphuric acid–ammonia ion-induced nucleation can trigger new particle formation in the winter time sub-arctic / boreal environment with a high level of anthropogenic $SO_2$ pollution but a low UV-radiation intensity. Data on neutral 1.5-3 nm particles are not available, so that neutral nucleation rates could not be derived. However, based on all the evidence obtained from the field (mainly Jokinen et al., 2018) and especially from the CLOUD experiments (Kirkby et al., 2011; Schobesberger et al., 2015), in the absence of significant amounts of compounds other than $H_2SO_4$ and $NH_3$, and with the nucleation rates being below the ion pair production rate (typically 2-4 ion pairs $cm^{-3}$ $s^{-1}$ in the Earth's surface layer), ion induced nucleation seems to dominate over the neutral one. In our case, HOMs were below the detection limit, and amines, if important, would appear also in the anion spectrum in $H_2SO_4$ clusters. $HIO_3$ and MSA were present, but significant neutral homogeneous nucleation of $HIO_3$ would require ~100-fold concentration of it compared to what was measured here (Sipilä et al., 2016).

The observation of clusters containing $IO_3^-$ or $HIO_3$ together with $H_2SO_4$ is, however, highly interesting. $HIO_3$ has been shown to nucleate homogeneously, and also mixed clusters containing both $HIO_3$ and $H_2SO_4$ have been reported from the Alps (Frege et al., 2017), Atlantic coast (Sipilä et al., 2016) and Arctic (Beck et al., 2021). If the $SO_2$-rich pollution plumes from the smelters are advected to iodine source areas (arctic ocean and especially sea ice zone as well as macroalgae rich coasts) or vice versa, this mixed mechanism may become important.

### 3.2.4 Particle growth and relevance for CCN-concentrations

Based on the above analysis, particle nucleation is clearly driven by sulphuric acid and ammonia, with the nucleation rate being most probably sensitive to concentrations of both of these vapours. But how do the freshly nucleated clusters grow? Assuming irreversible condensation, even the peak sulphuric acid concentration of $1.5 \times 10^6$ molecules $cm^3$ can explain only a small fraction of the observed growth rate. Consistent with an earlier report on wintertime particle growth rates at Värriö (Kyrö et al., 2014), the apparent average growth rate on 29 January was approximately 4.5 nm $h^{-1}$ (Fig. 2). Based on Stoltzenburg et al. (2020), such rate would require a steady $[H_2SO_4]$ of $2.6 \times 10^7$ molecules $cm^{-3}$ throughout the growth process which would continue long after the sunset when the $[H_2SO_4]$ already decays. Obviously, there are two possible explanations for this disagreement; either sulphuric acid was not responsible for most of the growth, or the air was not homogenous and the apparent growth was caused by the air mass advection (Kivekäs et al., 2016).

Besides sulphuric acid, the only condensable vapours detected were MSA and $HIO_3$ (and $NH_3$). However, their concentrations were clearly lower than that of sulphuric acid, and therefore condensation of those vapours in a homogeneous air mass cannot explain the apparent growth either. It could be speculated that compounds not recorded by the CI-APi-TOF,

such as $SO_2$ or some less oxidized volatile or semi-volatile organic compounds, (S)VOC, condense or react in particle phase forming low volatile compounds thereby contributing to growth (Stolzenburg et al., 2018). However, the complete absence of highly oxidized compounds does not support, though not fully exclude either, the presence of less oxidized compounds at a high abundance. The $NO_2$ concentration was moderate, up to 7 ppb, and therefore nitric acid concentrations were likely insufficient to have a measurable effect on the growth (Wang et al., 2020). However, the temperature was low during the studied time period, and therefore $HNO_3$ or some other semi-volatile compound could have contributed to the growth, provided that such compounds were present. Ammonia was detected in small ion clusters, but its contribution to the particle volume concentration, assuming that the measured cluster $NH_3/H_2SO_4$ ratio reflects the composition of larger particles, was marginal. Assuming the particle composition to be ammonium bisulphate, i.e. the $NH_3/H_2SO_4$ ratio of unity, ammonia would contribute 17% to the particle volume and 5% to particle diameter growth rate.

The most plausible explanation for the observed growth is that the particle growth was driven by $H_2SO_4$ condensation but its concentration was not uniform over the source area. In that case, particles would nucleate and grow to their final sizes during the few hours of sunlight. Particles formed and grown close to the emissions sources with high $SO_2$ and thus $H_2SO_4$ concentrations grow to larger sizes than particles formed near the measurement site. Air mass advection would then transport particles through the dark hours, leading to a steadily increasing nucleation (and later Aitken) mode diameters at SMEAR I, observed as an apparent steady growth even through the night. Modelling efforts and measurement of chemical composition or hygroscopicity of growing modes would be required for an unambiguous explanation of the particle growth.

New particle formation in the sub-Arctic winter would be irrelevant if formed particles would not grow to sizes (above few tens of nm) where they can act as CCN. We did not measure CCN concentrations at different supersaturations, but the air masses originating from the Murmansk–Kandalaksha region from about 03:00 onwards on 28 January (Fig. 6) contained elevated concentrations of Aitken and accumulation mode particles, mainly in the size range of ~20–500 nm (Fig. 10). New particle formation clearly increased the concentration of >3 nm particles, and also the concentration of particles larger than 50 nm showed an increase, especially on 29 January. The concentration of particles larger than 100 nm was relatively constant and apparently unaffected by NPF during the times when these NPF events could be observed. Air mass advection and particle loss processes, however, naturally have an impact on measured concentrations and are largely responsible for the development of particle populations.

Figure 11 presents the average particle number size distribution during about the one-week period of eastern winds (28 January – 3 February 2020), when the two clear NPF events presented above occurred. Concentrations of particles in all the size classes were remarkably higher, even by an order of magnitude for the 10-200 nm particles, than the average

concentrations during the preceding and succeeding periods with westerly winds. Concentrations during that one-week period were also clearly higher than the average concentrations between 1 November and 29 February, suggesting that new particle formation may be a significant source of particles in eastern air masses. However, primary emissions from the smelters and the surrounding cities would naturally show up in the size distribution plot as well. A more thorough analysis is needed to separate the roles of secondary NPF and primary emissions in the aerosol and CCN budgets. March, with almost continual NPF, was excluded from this analysis since the light conditions in March differ significantly from those between early November and end of February.

For an accurate assessment of contribution of secondary aerosol formation to CCN concentrations at SMEAR I or regionally, the meteorological situation, including boundary layer dynamics, wet deposition of particles, etc. should be considered. However, our observations on clearly elevated CCN-size aerosol particle concentrations in eastern air masses (Figs. 10 and 11) point towards a clear contribution of Kola Peninsula $SO_2$ emissions to winter-time CCN concentrations in the region.

## 4 Conclusions

Winter-time new particle formation and growth was investigated at the SMEAR I station, in Värriö strict nature reserve, Finnish Eastern Lapland. Sulphur dioxide concentrations in the air masses arriving from Kola Peninsula were often very high, occasionally over 30 ppb. At such high concentrations, enough sulphuric acid was formed to initiate new particle formation and growth, even in the presence of a very low solar radiation intensity.

New particle formation was observed mostly, but not solely, with easterly winds and in air masses arriving from the direction of Kola Peninsula smelters. Newly formed (4–10 nm, concentration $> 50$ cm$^{-3}$) particles were observed in altogether 34 days between 1 November 2019 and 15 March 2020, and out of these days about. 60% were associated with eastern winds or with the period of wind direction change from ~west to east. Nucleation was observed *in situ* at the SMEAR I station at $H_2SO_4$ concentration exceeding $1 \times 10^6$ molecules cm$^{-3}$. These cases were identified based on the appearance of ~1.5–2 nm ion clusters. Other NPF events were observed as appearances of particles of a few nm in diameter, and these particles gradually grew in size over time. Nucleation at SMEAR I was shown to proceed via a negative ion-induced sulphuric acid – ammonia (– water) channel which, based on prior understanding from laboratory experiments, can be hypothesized to dominate the NPF process at our site. Closer to $SO_2$ emission sources where $H_2SO_4$ concentrations are likely remarkably higher, nucleation can proceed also via a neutral channel and could, theoretically, involve compounds other than $H_2SO_4$, $NH_3$ and water.

Larger particles with a diameter of at least a few nm observed at SMEAR I, were probably not formed in the immediate vicinity of the site, so they had grown in size during the air mass advection. Secondary aerosol formation associated with the emissions from the Kola Peninsula emissions together with primary particle emissions impact the aerosol number size distribution, clearly increasing the concentrations of particles in all the size classes, and therefore unavoidably also CCN concentrations. For a better understanding of the contribution of $SO_2$ emissions from the Kola Peninsula to local and regional CCN concentrations, and for upscaling our results to cover the whole (sub)-arctic Eurasia with vastly polluting industrial cities such as Norilsk, require more measurements. Such measurements should be complemented with CCN or cloud residual measurements – ideally in more than only one location (SMEAR I) around the Kola Peninsula. Regional chemical transport and aerosol dynamic modeling would be necessary for a thorough assessment.

*Data availability*. Mass spectrometer data related to this article are available from Zenodo (https://zenodo.org/record/5524857#.YUyVoGYzbwc) as well as upon request to the corresponding author. Rest of the data are available for download from https://smear.avaa.csc.fi/.

*Supplement*. The supplement related to this article is available online at:

*Author contributions*. MS designed the experiment, MS, NS, KN, TL, DK, SK, LB, TL, JL, PPA, PK, ES, PAR, and TJ prepared the instruments, performed calibrations, collected the data and processed the data, MS and NS analyzed the data, EMD calculated the back trajectories. MS wrote the manuscript. All authors contributed to the interpretation of data and commented on the manuscript.

*Competing interests*. The authors declare that they have no conflict of interest.

*Acknowledgements*. We thank GiGAS-UHEL calibration centre for at-site CI-APi-TOF calibration, Värriö research station staff for technical support and Lubna Dada for discussions related to sulphuric acid calculation.

*Financial support*. This work was supported by the Academy of Finland (projects: 296628, 328290, 310627, 334514), the European Research Council (ERC) under the European Union's Horizon 2020 research and innovation programme (GASPARCON, grant agreement no. 714621) and the European Commission under the European Union's Horizon 2020 research and innovation programme (H2020-INFRADEV-2019-2, ACTRIS-IMP, grant Agreement no. 871115).

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

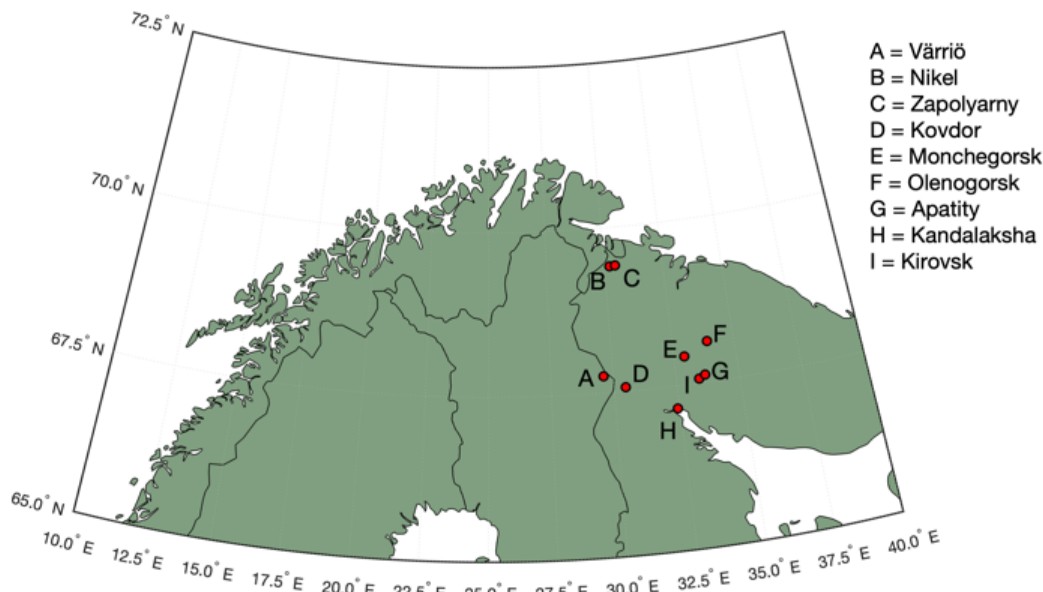

**Figure 1.** Map of the study area. Our measurement site is located in Värriö. Industrial cities of Nikel, Zapoljarnij, Monchegorsk, Kandalaksha have large-scale metal smelters emitting vast quantities of SO$_2$ into the atmosphere. Kovdor and Olenegorsk mines produce nickel/iron ore, but have no smelter industry. Kirovsk and Apatity are phosphate mining and processing sites.

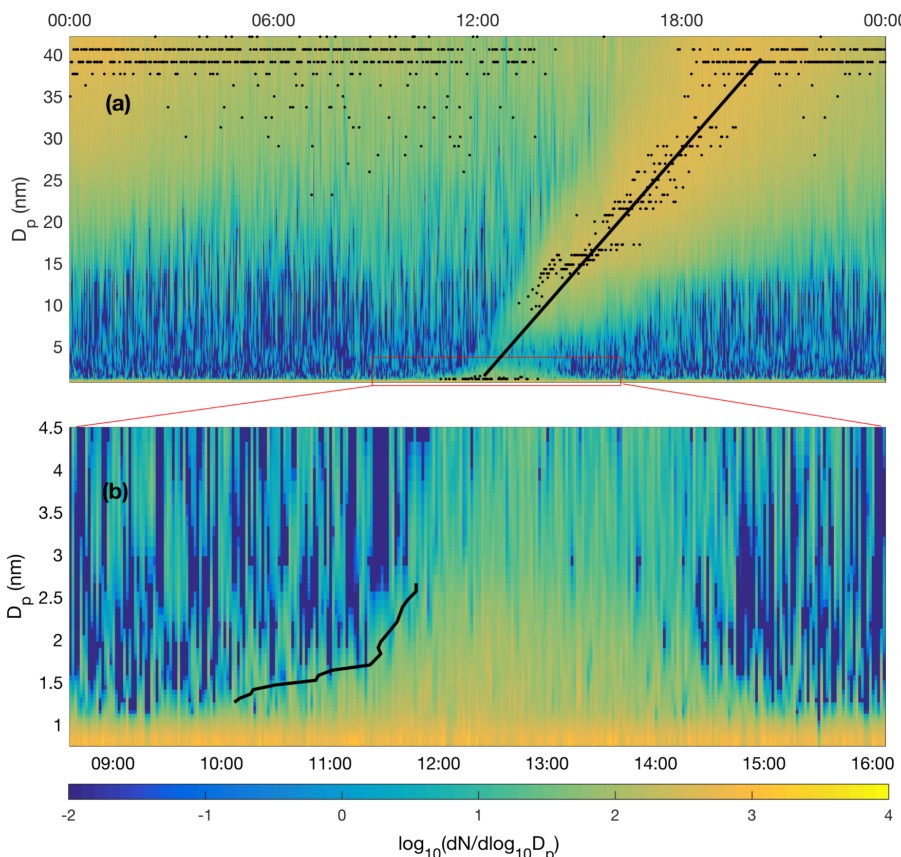

**Figure 2.** Particle formation event recorded by the NAIS (negative ions) on 29 January 2020, depicted on a linear diameter scale. The average growth rate determination by a linear fitting to the growing nucleation mode yields an average growth rate of 5 nm h$^{-1}$ (a) while the 50-% appearance time method (Lehtipalo et al., 2014) applied in cluster mode growth yields growth rates from about 0.35 to 1.8 nm h$^{-1}$ (b). Both methods likely overestimate true growth rate as the particle size distribution is affected by air mass advection. Therefore, the growth rate applied in nucleation rate calculations is derived from sulphuric acid concentration (Stolzenburg et al., 2020).

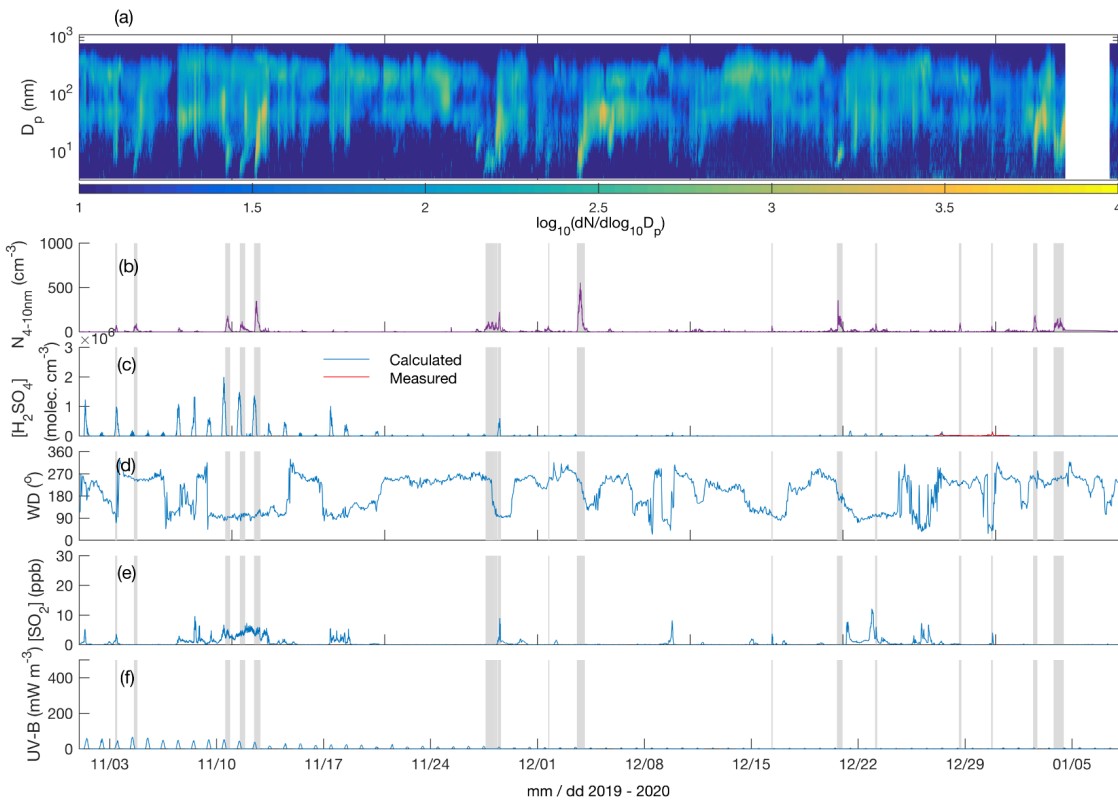

**Figure 3.** Particle number size distribution in the diameter range 4–700 nm (a), number concentration of 4-10 nm particles (b), measured (only between 27 and 31 December 2019) and calculated (Dada et al., 2020) sulphuric acid concentration (c), wind direction (d), $SO_2$ concentration (e) and UV-B radiation (f) over the time period 1 November 2019 – 07 January 2020. The gray shaded areas depict the times with observed < 10 nm new particle formation. The polar night (sun constantly below horizon) period is from 9 December 2019 to 4 January 2020.

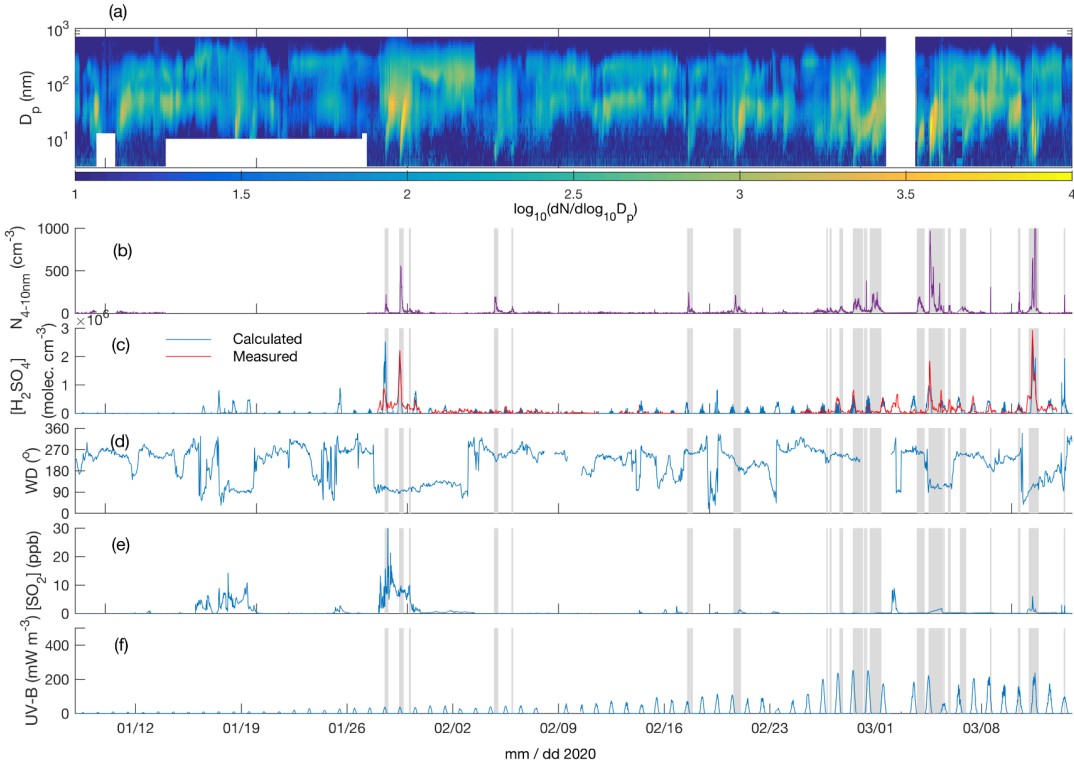

**Figure 4.** Particle number size distribution in the diameter range 4–700 nm (a), number concentration of 4-10 nm particles (b), measured and calculated (Dada et al., 2020) sulphuric acid concentration (c), wind direction (d), SO₂ concentration (e) and UV-B radiation (f) over the time period 8 January 2020 – 15 March 2020. The gray shaded areas depict the times with observed < 10 nm new particle formation.

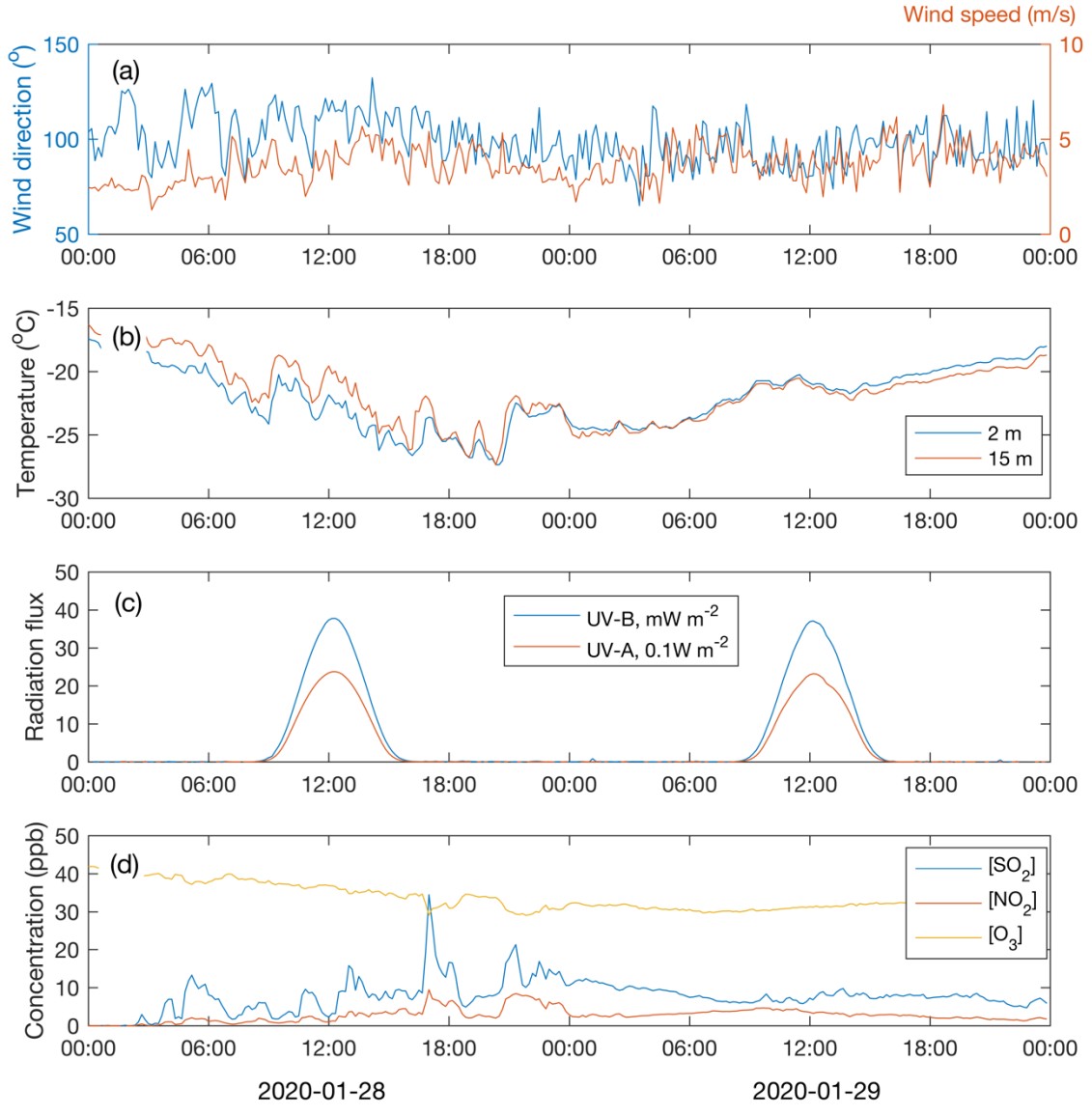

**Figure 5.** Wind speed and direction at the 16-m height (a), air temperature at two heights (b) , UV-B and UVA radiation (c) and concentrations of SO₂, NO₂, O₃ (d) during the period 28–29 January 2020.

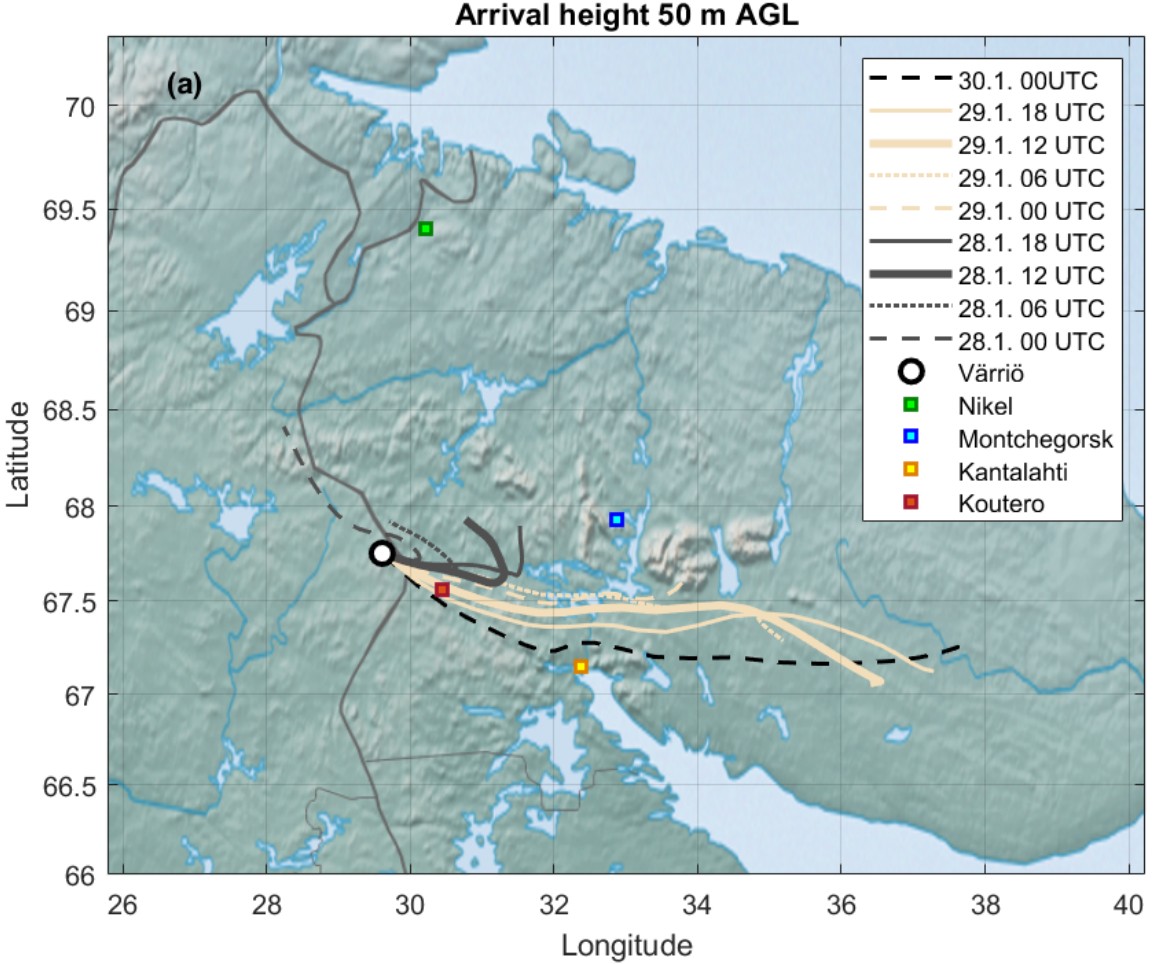

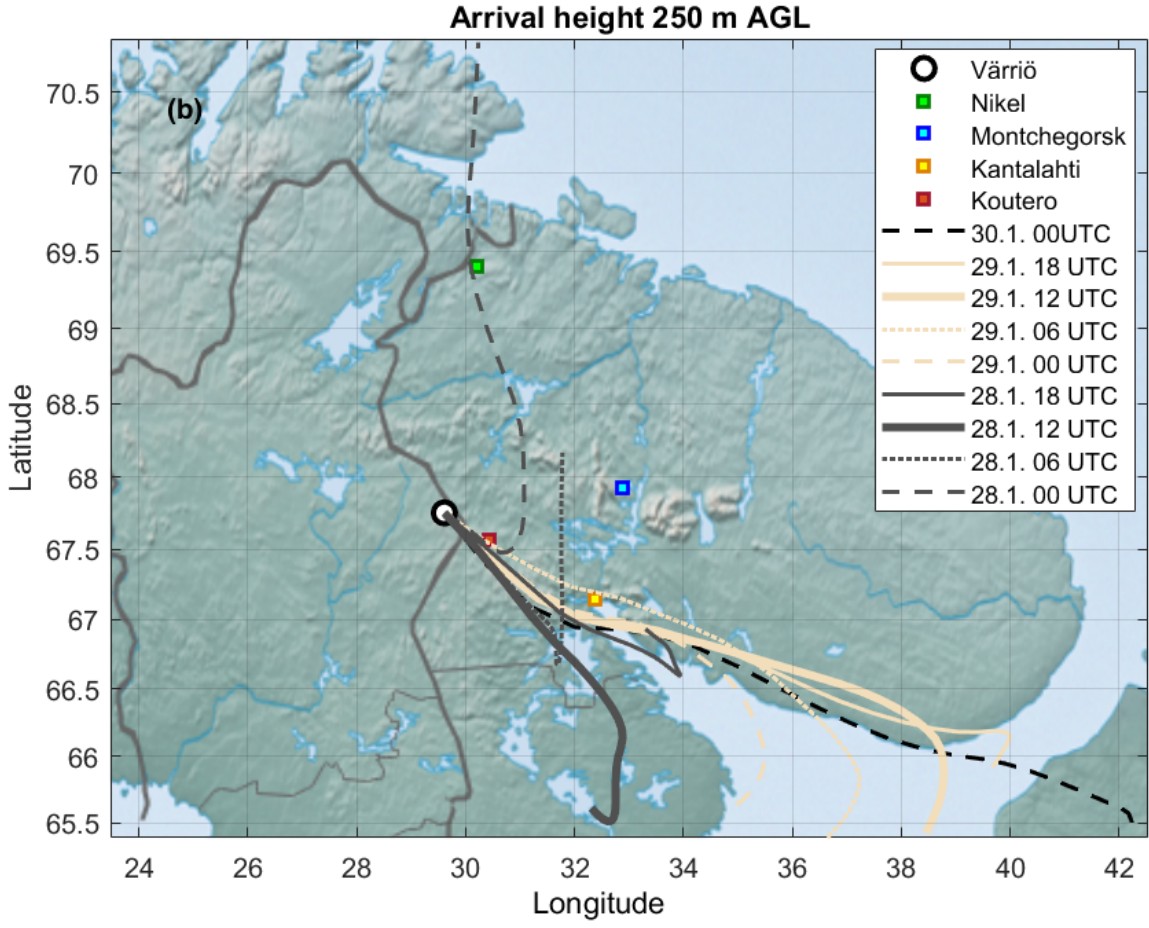

**Figure 6.** 24-hour back trajectories with an arrival height of 50 m (a) and 250 m (b) above ground level (AGL), and with arrival times between 28 January at 00:00 and 30 January at 00:00.

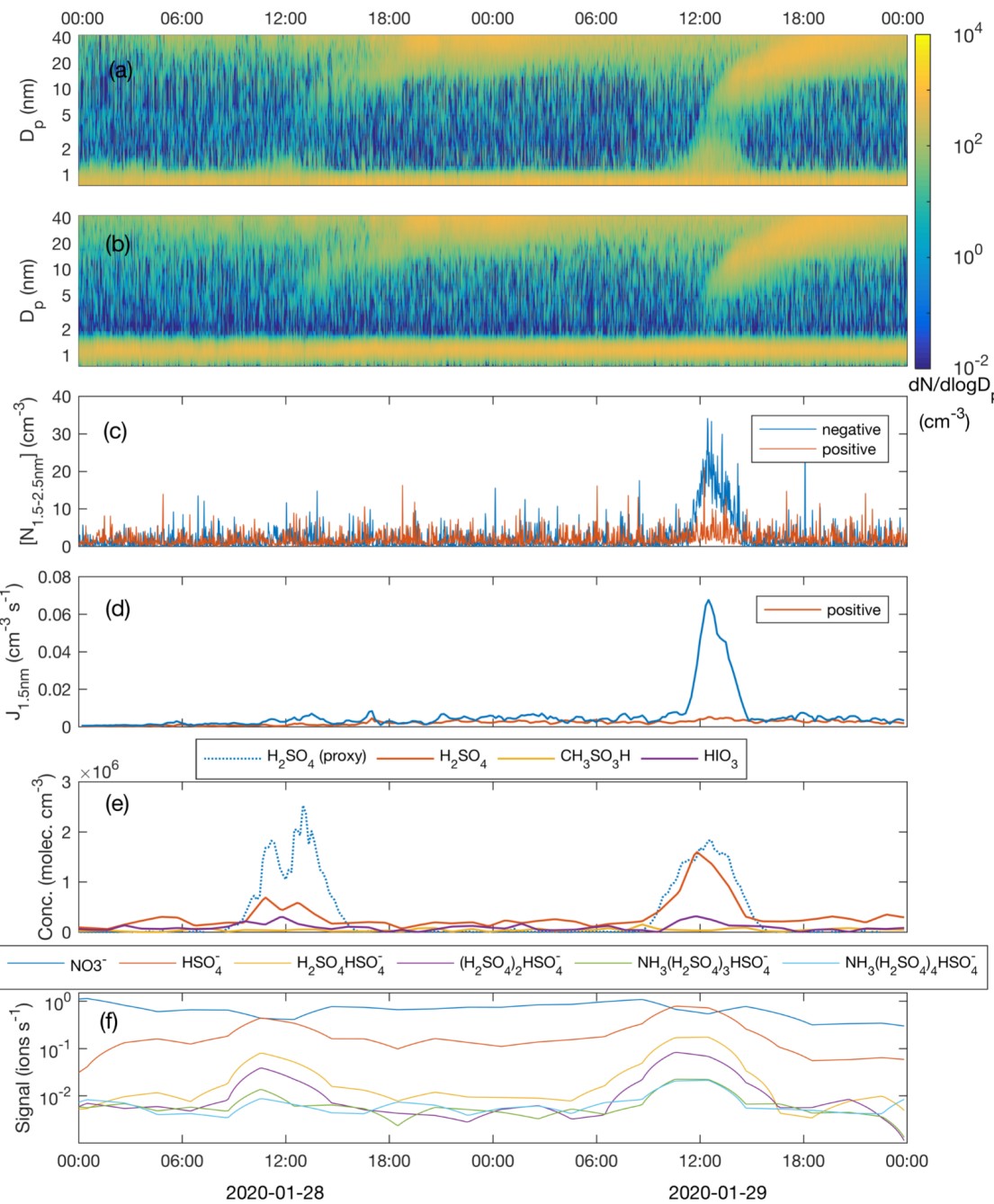

**Figure 7.** Number size distribution of negative (a) and positive (b) clusters and particles, concentration of freshly nucleated, charged 1.5–2.5 nm clusters (c), formation rate of negative and positive 1.5 nm clusters (d), measured concentrations of sulphuric acid ($H_2SO_4$), methane sulphonic acid ($CH_3SO_3H$) and iodic acid ($HIO_3$) as well as sulphuric acid concentration estimated by proxy calculation (e), and the signal intensity of nucleating ion clusters with different composition (f) during the period 28–29 January 2020.

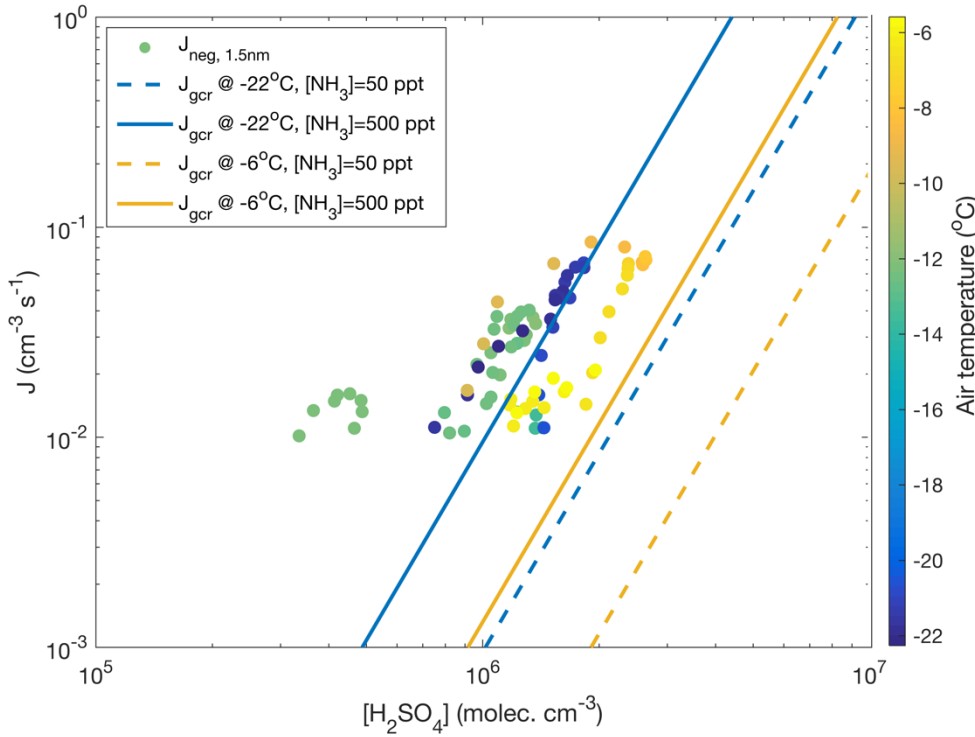

**Figure 8.** 1-hour average negative ion-induced nucleation rates calculated vs. calculated sulphuric acid concentration (Dada et al., 2020) for days with visible and clear nucleation events (11–12 November 2019, 18–19 November 2019, 28–29 January 2020, 13 March 2020) colored according to the air temperature. A nucleation rate of $10^{-2}$ cm$^{-3}$ s$^{-1}$ was used as a threshold for reliable determination below which instrument noise becomes predominant. No positive ion-induced nucleation was observed. For comparison, total (negative, positive and neutral) nucleation rates $J_{gcr}$ calculated based on CLOUD parameterization (Dunne et al., 2016) are presented. The calculation assumes a ground-level galactic cosmic ray ionization rate of 1.8 ion pairs cm$^{-3}$ s$^{-1}$ and no contribution from terrestrial radioactivity. The calculation was performed at –22°C and at –6°C assuming an ammonia concentration of either 50 or 500 ppt.

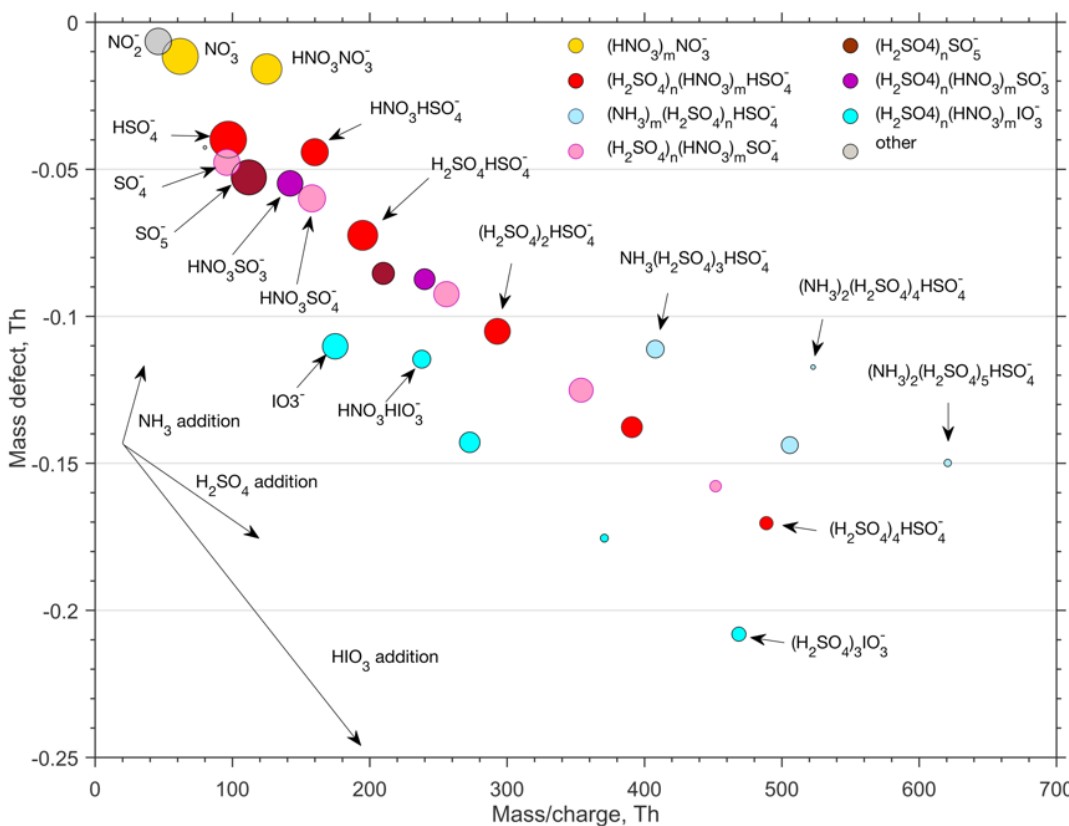

**Figure 9.** Mass defect plot (with a 2-h effective integration time) of the anion cluster distribution recorded by the APi-TOF during intensive cluster formation on 29 January 2020. The size of the circles is proportional to the concentration. See text for a detailed description.

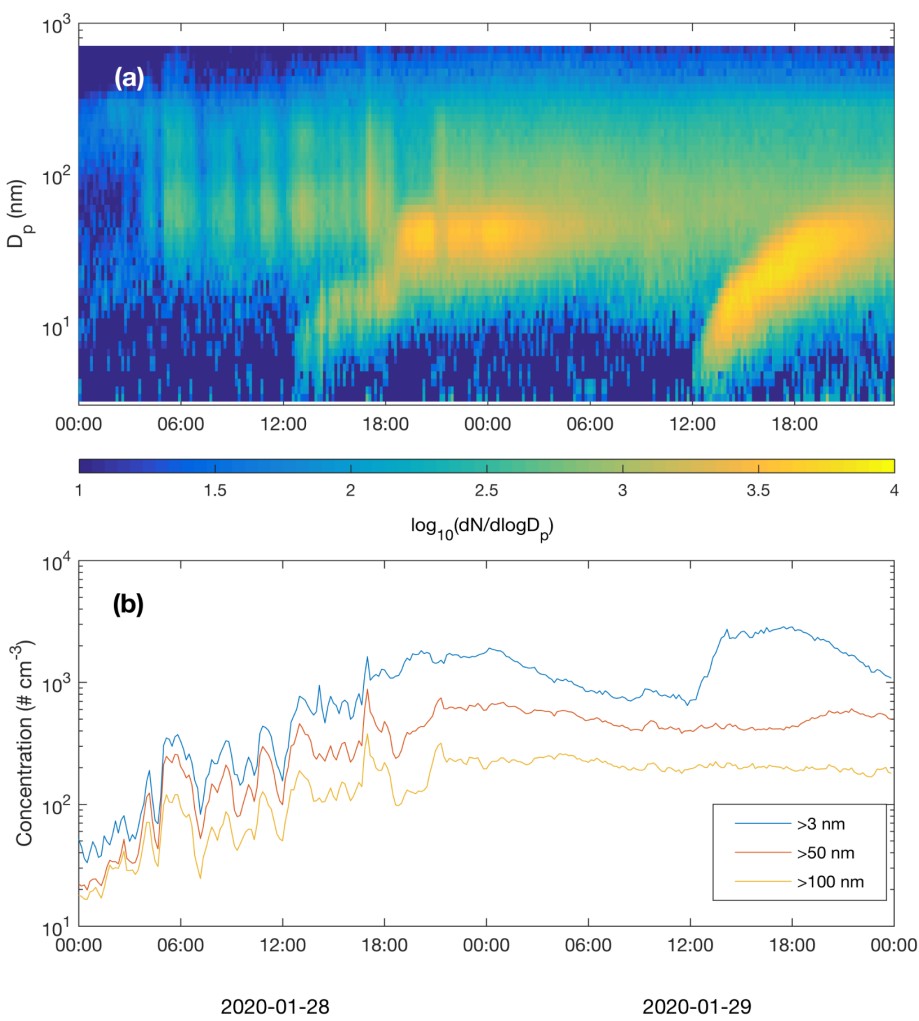

**Figure 10.** Particle number size distribution (upper panel) and concentrations of particles larger than 3 nm, 50 nm and 100 nm (lower panel) recorded by the DMPS.

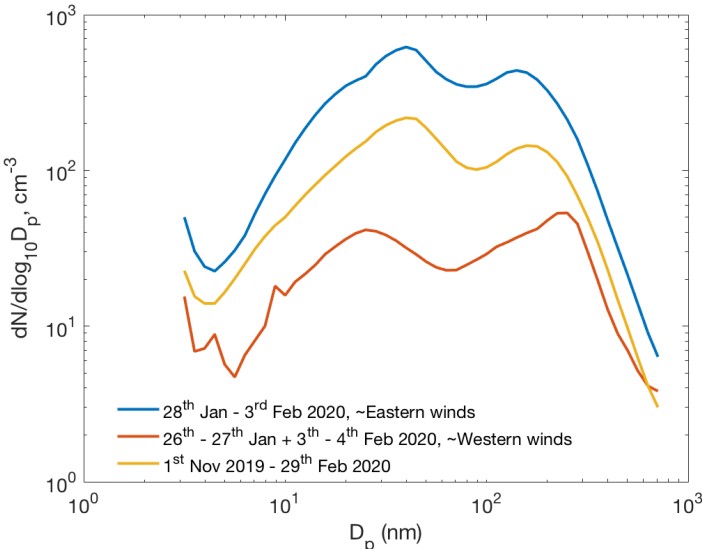

**Figure 11.** Average particle number size distributions during the ~one-week period of easterly winds (28 January – 3 February 2020), during the preceding and succeeding time period with westerly winds, and average number size distribution between 1 November and 29 February.