# Peer review of "Wintertime sub-arctic new particle formation from Kola Peninsula sulphur emissions"

_Atmospheric Chemistry and Physics, 2020_

## Author Response (AR1)

**Final response to referee #1 and #2 comments:**

We thank both referees for their thorough assessment and good suggestions to improve our MS. **We have accounted many of the suggestions, but due to lack of resources we cannot conduct very extensive additional data analyses** (though certainly conducted all that was doable with reasonable effort). Therefore, **this study remains as a case study -based investigation on winter-time new particle formation mechanism**. However, in our future work we will focus on a multi-decade analysis of NPF and its seasonal (including natural biogenic contributions), annual and decadal variation to get comprehensive view on the overall role of anthropogenic sulfur emissions in aerosol and CCN budgets in the area, and we wish patience from the referees, the editor and ACP readers to wait for us to accomplish those major investigations in the near future. **Below answers to referee comments and changes made for the MS. Major changes in MS text are highlighted with red.**

*Referee #1*

**General comment**

**The manuscript provides an interesting perspective on the effect of anthropogenic emissions for wintertime new particle formation (NPF) in the Arctic. In particular, Sipilä et al. show that the high SO2 emissions from industrial activities in the Kola peninsula (Russia) can lead to sufficiently high sulfuric acid concentrations to promote NPF despite the low amount of solar radiation. Previous studies[1] have already linked high level of SO2 from the Kola peninsula with NPF occurrence but this is the first study that quantifies sulfuric acid concentration and provides a molecular level characterization of the nucleating clusters.**

**The analysis is generally sound and the results are nicely presented. However, the manuscript focuses too much on a single case study. A more comprehensive analysis of the entire campaign would be highly beneficial to better understand the impact of industrial emissions on NPF compared to other processes. The authors, could for example look at the effect of SO2/H2SO4 and/or wind direction on the occurrence of NPF events. I also recommend to include a table listing all NPF events with the most relevant variables (e.g. wind direction, H2SO4 and SO2 concentration, J rate if available), this could be a very useful reference for future studies.**

The MS describes data from the first concurrent deployment of CI-APi-TOF and aerosol and ion cluster detectors from the site, and unfortunately the number of days with all instruments simultaneously operational is still quite sparse. Utilizing the limited amount of data presented here, would not give a exact picture on industrial emission and their impact on NPF in the area. Currently, we are collecting long term data and a comprehensive analysis of NPF throughout all seasons and over several decades is a topic of our future study. Unfortunately, that analysis is far beyond what we could accomplish in reasonable time to complement the present MS.

Here we to demonstrate that the physical NPF mechanism (ion-induced sulfuric – acid (-ammonia) nucleation) can and does occur. Amount of data providing direct evidence on nucleation mechanisms from field conditions is still extremely rare and ion-induced sulfuric acid – ammonia nucleation mechanism has so far demonstrated to *dominate* secondary new particle formation only in two prior studies (Jokinen et al., Science Advances, 2018; Beck et al., Geophys. Res. Lett. 2021). Therefore, we believe that showing the production of CCN-sized aerosol in the winter twilight even takes place, and resolving the nucleation mechanism is sufficient and serves as a basis for future's more comprehensive statistical assessments.

Problem with tabulating the J values is that the J's change nonlinearly with SA concentration. In earlier times when J's were back-calculated (e.g. Sihto et al., Atmos. Chem. Phys., 6, 4079-4091, 2006; Riipinen et al., Atmos. Chem. Phys. 7,1899-1914, 2007) using e.g. the approach by Kerminen & Kulmala (J. Aerosol Sci. 33, 609-, 2002), tabulation made more sense, since those analyses yielded some sort of representative J for the investigated event day. Here, where we measure nucleation (or, more accurately, we measure ion cluster

concentration changes in defined size range) directly we can obtain much better temporal time resolution, but tabulating that is not anymore a good way to represent those data. Instead, to provide better insight on relation between J and [H2SO4] we made a new figure (**Figure 8** in revised MS and also pasted below) showing the dependency J vs. H2SO4 in different temperatures. We used data from altogether 8 days with clear ion-induced nucleation including 28.-29.1.2020 presented in the main text, 10.-12.11.2019 and 11.3.2020 presented in Supplement and 18.-19.11. which is not separately presented. Data with no clearly observable nucleation was not included in the analysis. To guide the eye (rather than prove or disprove the mechanism) we included 4 lines to the figure. Those are total nucleation rates in 2 different temperatures and 2 different ammonia concentrations, calculated based on CLOUD-parameterization by Dunne et al. (2016). Our J's exceed slightly those calculated under our assumptions (see Figure caption) but, considering that they do not account for airmass mixing etc., difference is not very large. (Naturally that parameterization would be ideally applied in some sort of chemical transport model possibly resulting in better agreement.)

Text describing the result and figure was added to the MS chapter **"3.2.3 New particle formation"**, sub chapter "*Nucleation rates*", **lines 301-310** in the revised MS:

"Though this value is subject to a significant uncertainty, it would agree with observations of Kirkby et al. (2011) and parameterizations by Dunne et al. (2016) which yield a "slope" of approximately 3 under similar conditions as visualized in Figure 8. In the same figure, data from all days with clearly observable ion-induced nucleation are depicted. There, hourly average nucleation rates $J_{1.5^-}$ , that exceed a threshold value of $J_{1.5^-} = 0.01$ cm $^{-3}$ s$^{-1}$, are plotted as a function of concurrent calculated sulphuric acid concentration and air temperature. Calculated nucleation rates, $J_{GCR}$, represent the total nucleation rates (ion-induced plus neutral) in different temperature and ammonia concentrations under influence of galactic cosmic radiation (GCR) producing ions with the fixed rated of 1.8 ion pairs cm$^{-3}$ s$^{-1}$. Negative ion-induced nucleation, however, should be the dominant mechanism under these conditions (Kirkby et al., 2011) and results can be therefore compared. Our data is reasonably close to range predicted by parameterization considering that this simple calculation does not include airmass transportation, vertical mixing, terrestrial radiation sources, or any other real-world phenomena. Also, sources and concentration of ammonia in our study area are unknown. "

[Figure]

**Figure 8.** 1-hour average negative ion-induced nucleation rates calculated vs. calculated sulphuric acid concentration (Dada et al., 2020) for days with visible and clear nucleation events (11.-12.11.2019, 18.-19.11.2019, 28.-29.1.2020, 13.3.2020) colored according to air temperature. Nucleation rate of $10^{-2}$ cm$^{-3}$s$^{-1}$ was used as a threshold for reliable determination below which instrument noise becomes predominant. No positive ion-induced nucleation was observed. For comparison, total (negative, positive and neutral) nucleation rates $J_{gcr}$ calculated based on CLOUD parameterization (Dunne et al., 2016) are presented. Calculation assumes a ground-level galactic cosmic ray ionization rate of 1.8 ion pairs cm$^{-3}$ s$^{-3}$ and no contribution from terrestrial radioactivity due to thick snow cover. Calculation was performed at -22°C and at -6°C assuming 50 or 500 ppt ammonia concentration. Despite the fact that in these conditions the total $J_{gcr}$ is approximately equal to negative ion-induced $J_{neg}$, our J values slightly exceed those that would be expected based on parameterization assuming ammonia concentration does not exceed 500 ppt. However, the disagreement is such small for atmospheric nucleation rate measurement (which does not account for airmass mixing and transportation) that it should not be considered to conflict with the proposed mechanism.

**The manuscript would also benefit from a careful proofreading.**

Done.

**Specific comments**

**Nucleation rate calculation: I am concerned about the application of Stolzenburg[2] equation to estimate GR2. This equation was developed for the growth of neutral particles but you are using it for ions. The growth of charged nano-particles can be significantly faster because of the dipole moments of sulfuric acid [2, 3]. I understand your concern about fitting the entire PSD as shown in Fig.S1, which can lead to an overestimation of the particle growth rate due to airmass advection.**

Stolzenburg et al. (2020) found enhancement factor of 1.45 resulting from charge – dipole interactions and we thank referee for observing that we did not implement that. We recalculated the nucleation rates accounting for this additional growth factor and changed the text and equation:

$$GR_2 = 1.45 \cdot \left( 2.68 \cdot \left( \frac{d_p}{\text{nm}} \right)^{-1.27} + 0.81 \right) \cdot [H_2SO_4] \cdot 10^{-7} \text{molec.}^{-1} \text{ cm}^3 \, , \qquad (3)$$

where the pre-factor 1.45 accounts for dipole - charge interaction in charged particle growth (Stolzenburg et al., 2020)

**A possible solution could be to apply the appearance time method to the growth of ions smaller than 3nm. The lifetime of these ions is comparable to sulfuric acid and I think they should not be affected much by air mass advection. The advantage of this method would also be that it does not rely on the assumption that sulfuric acid condensation is the sole responsible for growth.**

Appearance time method would work probably better in a chamber where relatively stable SA can be produced after which one can investigate the growth by following the appearance of particles (ion clusters in this case) in different size channels. That would yield one GR and one J per one [H2SO4] and as name says, is time-wise limited to "appearance "of those clusters, though nucleation in our case continues despite the apparent "disappearance" (decay) of the cluster population after peak J.

Our [H2SO4] spikes quite rapidly around the noon and is definitely not stable. J is strongly H2SO4 dependent and if we want to plot J vs. time we should have time dependent GR. Using SA as GR proxy is the only way to do that. And since APi-TOF data shows, sulfuric acid is the primary component in the clusters, sulfuric acid condensation as growing mechanism is a good assumption.

However, we evaluated the 50-% appearance time for clusters in 1.3-2.7 nm size interval on 29[th] January. Result is presented in new **Figure 2b** (pasted below). Growth rate during the early event is approximately 0.35 nm h[-1] and seem to increase up to 1.8 nm h[-1] by the time, until the appearance method does not work anymore (~11:50). These values, together with the 4.5 nm h[-1] obtained by fitting to the total PSD are now applied in J calculation (Eq. 1) to test the effect of choice of GR to nucleation rate. Result is presented in new **Supplementary Figure S1** (see below). Manuscript text regarding growth rate (**2.3. Nucleation rate calculation** -chapter) was largely revised (pasted below). Conclusion is that derivation of J using our chosen sulphuric acid condensation based time-dependent J does not significantly differ from J derived using fixed appearance time GR. Application of GR from tit to PSD leads to more clear disagreement. New text:

"Rather than the average GR of whole particle population, the 50-% appearance time method (Lehtipalo et al. 2014) could be used to estimate the growth rate of nucleating clusters in the size range of 1.3 – 2.7 nm (Figure 2B). Here, the cluster appearance time in each size channel represent the time when cluster concentration reaches 50% of its maximum concentration during the event. Growth rate of can be assessed from cluster diameter vs. appearance time curve (black line) resulting in ~0.35 nm h[-1] between 10:10-11:30 and in ca. 1.8 nm h[-1] growth t 11:30-11:50 and the average growth rate of ~0.9 nm h[-1] during the period from 10:10-11:50. Drawback of this analysis is that the GR cannot be obtained for the period where concentration has passed the 50%-threshold or the period of decaying concentration. Furthermore, temporal GR cannot be properly obtained due to fluctuations in the data. Nevertheless, above mentioned values of 0.35 – 1.8 nm h [-1] can be compared to those obtained from Eq. 3, which yields maximum GR during the example day (29[th] January 2020) of 0.51 nm h [-1] around noon. Comparison to 1.8 nm h[-1] obtained from the 50%-appearance time method slightly before noon leads to a factor of ~3.5 difference in $GR_2$ on that day, which, in turn, is reflected in 22% difference in the calculated nucleation rate (Eq. 1). Effect of different determination of GR are visualized in Figure S1. To conclude, ion-induced nucleation rate calculation is not very sensitive to $GR_2$ because ion-ion recombination term (Eq. 1) dominates the loss in our conditions. "

[Figure]

**Figure 2. Particle formation event recorded by NAIS (negative ions) on 29. January 2020 depicted on linear diameter scale. Average growth rate determination by linear fitting to growing nucleation mode yields the average growth rate of 5 nm h$^{-1}$ (upper panel) while 50-% appearance time method (Lehtipalo et al., 2014) applied in cluster mode growth yields growth rates from ca. 0.35 to 1.8 nm h$^{-1}$ (lower panel). Both methods likely overestimate true growth rate since particle size distribution is affected by air mass advection. Therefore, the growth rate applied in nucleation rate calculations is derived from sulphuric acid concentration (Stolzenburg et al., 2020).**

[Figure]

**Figure S1. Nucleation rate (J$_{1.5nm}$) on 29$^{th}$ January 2020 calculated by Eq. 1 using growth rate (GR) calculated from Eq. 3 that assumes irreversible sulphuric acid condensation as sole mechanism of growth (Stolzenburg et al., 2020) (blue line), lower**

**(magenta) and upper (black) limits for GR derived from cluster 50-% appearance time (Lehtipalo et al., 2014) and GR derived from fitting to total particle size distribution (red).**

**Ion induced nucleation (IIN): you often refer to IIN as the driving NPF mechanism. However, without a measurement of neutral particle formation rate is not really possible to say if IIN is the dominant mechanism. I agree that, based on previous experiments/field observations, IIN would probably play a major role given the low sulfuric acid concentrations. However, you should state more clearly and earlier in the text that this is just an hypothesis.**

Added "However, due to lack of information on neutral ~1.5nm – 3 nm cluster concentrations this observation alone is not excluding parallel neutral nucleation mechanisms." **Lines 280-281.**

Conclusions read now: "Nucleation at SMEAR I was shown to proceed via a negative ion-induced sulphuric acid – ammonia (– water) channel which, based on prior understanding from laboratory experiments, can be hypothesized to dominate the NPF process at our site. Closer to $SO_2$ emission sources where $H_2SO_4$ concentrations are likely remarkably higher, nucleation can proceed also via neutral channel and could, theoretically, involve compounds other than $H_2SO_4$, $NH_3$ and water. "

New text related to response to earlier comment above also now adds to this by stating: "Negative ion-induced nucleation, however, should be the dominant mechanism under these conditions (Kirkby et al., 2011) and results can be therefore compared" approximately on **line 308.**

**Moreover, I am also wondering to which extent the low ion formation rates can explain the increase in the particle number for the event presented in the main text. In particular, you report a maximum formation rate of about 6E-2 ions/(cm3*s) (Fig.5) but the increase of particles larger than 3nm shown in Fig.7 is larger than 1E3 over the course of an hour roughly, which translates in a formation rate of about 0.3 particles/(cm3*s). This rate is 5 times faster compared to the ion formation rate and it is just a lower limit estimate (losses are not considered). Do you have an explanation for this? Maybe it would be beneficial to compare the ion formation rate with the formation of neutral particles larger than 3nm.**

Ion pair production rate only due to cosmic radiation is ~2 ions pairs cm-3 s-1. Therefore, ion production rate is not limiting factor and 0.3 vs. 2 allows also reasonable coagulation losses to take place between nucleation and growth (during airmass advection) to DMPS observable sizes. I.e. Nucleation rate closer to emission source just must be clearly higher (>0.3 cm-3 s-1) than at our site.

**Sulfuric acid proxy: I appreciated the application of the sulfuric acid proxy from Dada et al. 2020[4] to estimate the sulfuric acid concentration and it is nice to see the agreement on Fig.5. However, I would like to see a comparison between the sulfuric acid measurement and the proxy for the entire campaign when data are available. This comparison would be helpful to understand the applicability of this proxy to other Arctic locations where sulfuric acid measurements may not be available, providing a useful reference for the entire community. I was also intrigued by the hypothesis that the discrepancy between the proxy and the measurement on January 28 may be due to the strong surface inversion and it would be interesting to see how often this effect is present. If you do the comparison it should be easy to see if stronger surface inversions lead to higher discrepancies.**

CI-APi-TOF suffered from several malfunctioning periods earlier in the winter which is the cause for lack of data in early winter events reported in Supplement. We plotted all available data 25th Dec 2019 – 15th March 2020 in **Figures 3 and 4**. Despite inversion periods leading to more clear disagreement, the agreement is in general very good which we now discuss in the main text:

"During the times when CI-APi-TOF was operational, the agreement between the measured and calculated concentrations was good (Figure S2) with mean concentrations agreeing within 8%. Obtained correlation coefficient was R = 0.790 and coefficient of determination $R^2$ = 0.624."

And with the help of a new **Figure S2** in Supplement:

[Figure]

**Figure S2. On average, calculated (Dada et al., 2020) sulphuric acid concentrations agree very well with the measured sulphuric acid concentrations interpolated to same time axis. Difference between the mean concentration is only 8% (Average calculated concentration is 8% larger). This suggests that proxy by Dada et al. (2020) can appropriately describe the winter time sulphuric acid concentrations. Natural fluctuation in data is caused at least by meteorological phenomena such as boundary layer inversion. Here we used a CI-APi-TOF approximate lowest detection limit of $5 \cdot 10^4$ cm$^{-3}$ (Jokinen et al., 2012) as a threshold value both for measured and calculated concentration. Data comprises all times when both CI-APi-TOF data and data used for proxy calculation were available.**

Related to surface inversion, such strong inversion did not occur anymore after January 28. We'd need more data for that but in general the comparison of measured and modeled SA to observe surface inversions is an interesting idea and we thank referee for that.

**Particle growth: in the manuscript you show that that particle growth cannot be explained by species measured with the nitrate CIMS (i.e. sulfuric acid, MSA, iodic acid and HOMS) and conclude that the growth must be happening closer to the source. I think this hypothesis is plausible, however, you should mention that the growth could also be driven by condensation of organics which are not detected by the nitrate CIMS. Stolzenburg et al. 2018 [5] have shown very clearly that at -25C nano-particle growth is mainly driven by organics not detectable with a nitrate CIMS. I can imagine that the industrial activities in the Kola peninsula also lead to high organic emissions and these may very likely contribute to particle**

**growth. Unfortunately with your dataset is not possible to distinguish between these two possibilities but you should discuss both of them and probably leave the question on the growth mechanism open.**

In original MS we state that: "Either sulphuric acid is not responsible for most of the growth or the air is not homogenous and the apparent growth is caused by the airmass advection."

and that: "It could be speculated that compounds not recorded by CI-APi-TOF, such as $SO_2$ or some volatile or semi-volatile organic compounds, (S)VOC, react in particle phase forming low volatile compounds therefore contributing to growth but we have no evidence on such a process."

We modified the latter sentence a little and added Stoltzenburg et al., 2018 reference ". It could be speculated that compounds not recorded by CI-APi-TOF, such as $SO_2$ or some less oxidized volatile or semi-volatile organic compounds, (S)VOC, condense or react in particle phase forming low volatile compounds therefore contributing to growth (Stolzenburg et al., 2018) but a complete absence of highly oxidized compounds does not support the presence of less oxidized compound at a high abundancy." Lines 364-367.

In Abstract, we added few words, now it states: "… and other growth mechanisms and condensation of other compounds cannot be fully excluded."

**Minor comments**

**Line 12: I would mention the first source of air pollution in the Arctic** Ok
**Line 30: OECD is not defined**. Ok
**Line 30: "converged" you mean decreased?** Ok
**Line 46: you should mention that SO2 leads to sulphate accumulation in aerosol particles as well (most of the clouds do not precipitate).** Done, says now "…liquid phase in cloud droplets, which may evaporate leading to sulfate aerosol production or precipitate as acid rain"

**Lines 72-83: Include also nitric acid nucleation.** Done, no we say: "Recent laboratory studies that probed nucleation of nitric acid and ammonia suggest the mechanism may contribute to new particle formation and growth especially in the upper troposphere (Wang et al., 2020).

"

**Line 103: "another DMPS"? You mean one of the two DMPS?** Another -> The other
**Line 127: What is the frame size? Maybe you can remove this info as it is not very relevant** Removed
**Eq. 2: the notation with Dp/nm is confusing, I would use Dp [nm]** This equation is copied from original Stolzenburg - paper and since in same equation [ ] are used to indicate sulfuric acid concentration [nm] could cause confusion. As editor wishes.
**Line 198: What does a "reasonably strong event" mean?** something which is clear but not massive.. Removed "reasonably strong".
**Line233: there is a reference missing probably.** Done

**Lines 321-335: I like the application of Schobersberger et al. 2015[6] results but you should mention that fragmentation can have an effect on this comparison.**

Good point, and absolutely true. We rewrote this part:

"If true, that would indicate an ammonia concentration of the order of ~$10^7$ molecules cm$^{-3}$, or ~1 pptv. However, cluster fragmentation inside the mass spectrometer can be totally different between our experiment and Schobesberger et al. (2015) study and therefore any conclusions on ammonia concentration should not be drawn. Nevertheless, if $NH_3/H_2SO_4$ – ratio would be low, that would mean that the system is not saturated with respect to $NH_3$ and that the

nucleation rate should therefore be sensitive to both $H_2SO_4$ *and* $NH_3$ similar to Jokinen et al. (2018). This together with unknown effects of cluster fragmentation, highlight the importance of…" Lines 367-373 in revised MS.

**Line 348: Frege et al.[7] already reported the observations of HIO3-H2SO4 clusters, so I would not say that this has not been observed before**. Yes that is true. Also our recent paper (Beck et al., 2021) reports that. Fixed.

**Line 368: Something is missing in the sentence.** True, now it says "However, the temperature was low during the studied time period and therefore $HNO_3$ or some other semi-volatile compound could have contributed to the growth, provided that such compounds were present"

**Line 386: Do not start a sentence with >100**. Fixed.

**Line 399: The acidity of aerosol particles is not necessarily related with their hygroscopicity and anyway you don't have information on particle composition so I would just remove this sentence**. Well, ok, removed.

**Data availability: the link does not work and as stated by the ACP data policy all data should be deposited in a public repository. Please avoid using the "data are available upon request" sentence.**

Fixed the link and will deposit the rest of the data to Zenodo.

**The reference to Dada et al. 2020 in the bibliography is missing.** Fixed

**Figure 2: You should not use the Jet color map, can you switch to a perceptually uniform colormap (e.g. parula in matlab or viridis in python)?** Ok.

**Figure 4: The figure is not easy to read, for example there are not relevant information and some trajectories have the same color. Can you replot the Hysplit data instead of just copying the original output?** Replotted.

**Figure 5: The ion color scale is missing, also change the jet color map. Figure 7: Change the jet color map.** Ok

**References**

[1] Kyrö et al. 2014 https://doi.org/10.5194/acp-14-4383-2014
[2] Stolzenburg et al., 2020 https://doi.org/10.5194/acp-20-7359-2020

[3] Svensmark et al., 2017 https://doi.org/10.1038/s41467-017-02082-2

[4] Dada et al., 2020 https://doi.org/10.5194/acp-20-11747-2020
[5] Stolzenburg et al., 2018 https://doi.org/10.1073/pnas.1807604115

[6] Schobersberger et al. 2015 https://doi.org/10.5194/acp-15-55-2015

[7] Frege et al., 2017 doi:10.5194/acp-17-2613-2017

*Referee #2*

**General scientific comment:**

**The manuscript presents evidence on how important strong pollution sources emittting SO2 can influence or initiate nucleation events downwind of the sources in remote areas. The study is mostly based on case studies. One case study is presented in detail in the main manuscript. Three other case studies are presented in the supplemental part whereof in one case measured acids and ion clusters are missing, in a second one measured ion clusters are missing and in a third one measured ion clusters are missing and other explanations are listed for the respective NPF event.**

**The study delivers interesting results on pollution induced new particle formation, but some more details especially on statistics with respect to the whole measurement period are strongly recommended to include before final publication.**

**The manuscript misses a detailed overview to put the case studies in the context of the full measurement period which was a couple of month in the winter period. An overview of all nuclation events, with regard to levels of SO2 concentration, sulfuric acid concentration (measured and modelled), wind direction, available UV radiation…**

This was partly given in original figure 2 which was now separated into two separate **Figures 3 – 4** to improve readability. Measured SA concentration was now added to those figures now also. J vs [SA] relation is now highlighted in new **Figure 8** and calculated and measured SA concentrations are compared in new **Figure S2**. We believe that these amendments appropriately address the request.

**…quality of event (number of clusters and further growth, etc.) would place the events in a context which is needed the evaluate the abundance of natural and anthropogenic nucleation events during the measurement period.**

… We have very limited amount of data and as described in our response to other referee's comments, we currently are collecting data on continuous manner for more comprehensive future study.

This is a case study that shows that sulphuric acid – ammonia IIN can occur and dominate the secondary aerosol formation besides the two locations where it has shown to control the secondary aerosol concentrations previously – coastal Antarctica (Jokinen et al., 2018) and coastal Svalbard (Beck et al., 2021). Due to serious lack of field observations on molecular steps of NPF globally, we believe case study approach is justified and future work will focus on investigations on occurrence of the phenomenon over seasons, years and decades. **Figures 3 & 4** (split from original Figure 2) can be used to assess the frequency of events (which almost for sure require also nucleation to take place). We state in revised MS that: "Newly formed (4-10 nm, concentration > 50 cm$^{-3}$) particles were observed in altogether 34 days between 1$^{st}$ November 2019 and 15$^{th}$ March 2020 out of which ca. 60% were associated with eastern winds or with the period of wind direction change from ~west to east.".

There was indeed a small flaw (counting error in number of event days) in original MS, which we fixed. **Lines 457:459.**

We have, on purpose, limited our analysis to winter period, when no biogenic precursors (HOM) are yet detected and also any marine plankton blooming cannot be expected. Also MSA, which would be a proxy for pelagic plankton activity and which would indicate biogenic SO$_2$ and H$_2$SO$_4$ source and which, itself, could contribute to growth if present in reasonable concentrations is very low. So, we have no indication on any natural nucleation events, so the evaluation of those is straightforward. And most likely either all events

observed are anthropogenic or like few discussed in text, we don't have clear understanding..  We'll revise the text to make sure this is properly said.

**Right now the manusctipt delivers case studies within a larger measurement period and it is difficult to follow the significancee of the findings. What kind of types of NPF was observed in March?**

As the spring proceeds the solar radiation intensifies and produces more sulfuric acid. That is reflected to increasing event activity. Despite the lack of ion-cluster data on e.g. 11th March, only sulphuric acid pops out in the precursor spectrum so nothing suggests there would be something fundamentally different in comparison to e.g. $28^{th}$ – $29^{th}$ January events in Figs 5 – 6.

In the end of March / early April  onwards (not shown) the biogenic production starts which results in increased concentrations of HOM. These HOM then will contribute to NPF during summer season. These summertime formation events will be analyzed and included in our future study, but cannot be included here. Therefore, we only report wintertime data and end our analysis in mid-March when HOM do not yet show up.

**Were there periods when conditions for NPF were favourable based on pollutant emissions, but NPF did not take place?**

No, not really, see our reply to same comment below.

**In addition, the paragraph on growth of observed particles and their contribution to CCN remains a bit vague and needs to be elaborated on.**

See our response on multiple later comments regarding growth.

**Figure S1 shall be included in the main manuscript.**

Ok. Now it is. (Fig 2a)

**The manuscript requires a native speaker to improve the language!**

Language has been improved.

**Detailed scientific comments:**

**Abstract**

-

**Introduction**

Page 1, line 25-26:

Comment: I would say that SO2 contributes to acidification of ... by atmospheric aerosol and cloud formation ... (check the sense of the sentence).

Page 2, line 32-33: Comment: Can you give a reference here?

Page 2, line 45-47: Comment : Check syntax of the sentence!

Page 3, line 68: Better: ... was published by ... Page 3, line 89:

Comment: 300 km is not so close in terms of distance between the observations and the emissions. I recommend to write the disctance explicitly here and also at other places in the manuscript (see abstract). 300 km gives some time for transport and corresponding processing!

**2 Methods**

**2.1 Site and time of the study**

-

**2.2 Instrumentation**

Pag4, line 103:

The part of the DMPS for ultrafine particle detection malfunctioned ...

Pag4, line 109-114:

The paragraph is misunderstanding, write exactly when this instrument was used and since when with the switcher, etc.

**2.3 Nucleation rate calculation**

-

**2.4 Sulfuric acid proxy calculation**

**This whole section leaves some questions. Because of missing data, the authors make a number of assumptions on the sulfuric acid concentration. Sulfuric acid is calculated using SO2 oxidation by OH which is proxied by global radiation; Criegee Intermediates proxied by monoterpene and ozone concentrations, condensation on pre-existing aerosol, assumption on monoterpenes, and global radiation assumption was used. This needs to be verified and some general evaluation of this method using the campaign dataset should be discussed.**

We plotted all available measured SA data (25th Dec 2019 – 15th March 2020) **in Figure 3 and 4** (Originally Figure 2, which was split in two). Despite inversion periods leading to more clear disagreement, the agreement between measured and calculated SA is in general very good which we now discuss in the main text :

During the times when CI-APi-TOF was operational, the agreement between the measured and calculated concentrations was good (Figure S2) with mean concentrations agreeing within 8%. Obtained correlation coefficient is R = 0.790 and coefficient of determination $R^2 = 0.624$. Lines:

And with the help of a new **Figure S2** in Supplement and also shown above in our response to referee #1.

**2.5 Trajectory analysis**

-

**3 Results and Discussion**

**3.1 New particle formation during the measurement period**

**General comment: As the study is mainly based on case studies, an overview table of these studies is needed stating the differences and similarities of the different events. The descrption here is otherwise confusing to evaluate which events follow certain patterns and which do not. What about situations where NPF would be expected to happen based on the general conditions, but it did not?**

Instead of a table we added the measured sulfuric acid concentration to the overview figure and splitted that in two to make it more readable. Also we added figure J vs SA, that shows the connection between J and SA. These amendments should address the need of better overview. We don't see the added value of a table, and what information it should contain? Concentrations and meteorological factors etc. are constantly changing during each day and each event, so we believe that a plot would serve the purpose better. We have ca 30 days with elevated small particle concentration indicative of NPF with numerous associated factors which are shown in **figure X & X**. Data have been / will be deposited to repositories and will be available for more detailed investigation. If requested, we could spilt the figure in even more separate figures but those likely should be represented in the Supplement. See our response to referee #1 also.

We really did not see cases when NPF was expected to happen, but did not. No cases with elvated SA and clearly lacking event were observed but every time when SA exceeded ~1e6 also NPF (presence of small sub-10 nm particles) was observed.

**Page 6, line 170 - 179:**

**Comment: Please redo the figure, it is not possible to see specific days because of the overall scaling of the time axis**

Figure was split in two to improve readability. We could consider also splitting it into still more separate plots and report those in supplementary if that is considered useful. On the other hand, all data shown in Figure have been (size distribution, meteorology, SO2 are in Smart-SMEAR service) / will be (measured sulfuric acid concentration goes to Zenodo asap) submitted to repository during the production of final paper and therefore reader interested in exact numerical values will readily have access to those data.

**It might be an idea to add boxes where events have taken place and label these boxes with the respective event dates.**

Grey shaded areas of Figs 3-4 depict the event periods (times with elevated sub-10nm concentration).

**Page 6, line 177: ... consequent days? ...**

-> consecutive.

**3.2 Case study 28th – 29th January 2020**

**3.2.1 Meteorological situation and trace gas concentrations** Page 7, line 204-205:

**Comment: It must be possible with Hysplit to calculate boundary layer height aalong the trajectory. This is very useful to see that air masses even at low altitudes were not above the boundary layer height and air was not mixed in from above.**

We reran the Hysplit analysis and made new plots (Figure 6 a&b) with 50m and 250m arrival height and modified the text:

"Trajectories were calculated by using the Hybrid Single-Particle Lagrangian Integrated Trajectory model HYSPLIT (Stein et al., 2015) with GFS 0.25 degree meteorology as an input. We calculated 24-hour backward trajectories arriving at 50 and 250 meters above ground level for the period 28.1.2020 00UTC to 30.1.2020 00 UTC, arriving every 6 hours. The trajectory calculations included mixing layer depth along the trajectory." and .

**3.2.2 Aerosol precursors**

**Page 8, line 224:**

**Comment: Do you mean a gradient in temperature per meter?**

Fixed. m-2 -> m-1

**Page 8, line 233: Can you add a reference here?**

Done

**3.2.3 New particle formation Page 9, line 264:**

**The scale in the graph only shows up to 0.06 while the full peak is observed? Please correct the text here!**

Figure (that was corrected after adopting GR enhancement factor) was updated and text modified accordingly.

**Page 11, line 323-325: Comment: Check syntax of this sentence!**

Sentence has revised.

**Page 11, line 323: ... on that data ...**

Ok, so no plural "data".

**Page 12, line 345:... compared to what was measured here ...**

Ok.

**3.2.4 Particle growth and relevance for CCN-concentrations**

**General comment: This paragraph does not really give useful information on the availability of CCN related to pollution induced NPF events. There is a need for a more thorough analysis.**

We agree that more thorough analysis utilizing regional chemical transport modeling combined with aerosol dynamic modeling or something equivalent is needed. Also new measurements (hygroscopicity, CCN, cloud residual, etc,) are needed. This is said in the text "Modelling efforts and measurement of chemical composition or hygroscopicity of growing mode would be required for an unambiguous explanation of the particle growth." **Lines 420-421** of the revised MS.

Otherwise we slightly disagree with the referee and believe that our paragraph (and MS in general) give useful information on mechanism of formation, gives plausible mechanism for growth (and excludes many mechanisms involving HOM,

MSA, HIO3, HNO3, etc.) and also shows clear growth to CCN sizes. See our more comprehensive answer to the other comment about the same topic below.

**Page 12, line 365: Comment: Here a more thorough explanation for the potential particle growth is needed. A literature review on potential VOCs in a similar environment during similar seasons would be sufficient here.**

We now say with citation to Stolzenburg et al. (2018), that "It could be speculated that compounds not recorded by CI-APi-TOF, such as $SO_2$ or some less oxidized volatile or semi-volatile organic compounds, (S)VOC, condense or react in particle phase forming low volatile compounds thereby contributing to growth (Stolzenburg et al., 2018), but a complete absence of highly oxidized compounds does not support, though not fully exclude either, the presence of less oxidized compound at a high abundancy." We don't see the need for further literature reviews and I am not aware of anyone measured wintertime VOCs in our study area. Elsewhere / other times of year, there can be other VOCs with different emission rates, which would not add to our speculation about their role especially since nothing indicates their presence with significant levels. If we would have any indication on (S)VOCs, a review would be more justified.

**Page 12, line 367-368: Comment: Check the sentence on nitric acid, this does not make sense! Page 12, line 373-374**

Fixed

**Again, this explanation about the growth does not say anything.**

I'm not sure which lines referee now exactly refers to. Assuming we are still in the same paragraph with nitric acid, in my opinion the explanation says much. It says:

1) Condensation of MSA and HIO3 or HOM (the only known vapours that have been identified to grow nanosized particles besides SA) cannot explain the growth.
2) Further, it says that it could be speculated that (S)VOC makes up the growth, but we have absolutely no evidence on that.
3) Moreover, it states, that nitric acid is unlikely doing the job either and
4) even tells which fraction of growth ammonia would make if it would.

Then, because out of all condensable vapours, we only see sulfuric acid, the next paragraph suggests that

5) it most plausibly is sulfuric acid.

But in that case

6) Sulfuric acid is not homogeneously distributed.

I think that all these make total sense and tell quite a bit about growth beyond what has been shown earlier.

**You say that some concentrations were measured, but too small, but in the vicinity probably higher, explaining the process is absolutely vague and with no real evidence on the process.**

What (reasonably obtainable) evidence would be sufficient? We have the data what we have and cannot set up a matrix of measurement stations in Kola peninsula to measure the concentration gradient. In general, it is quite natural that after exiting the source plant, the emission dilutes.

We could leave the discussion away, but I believe that it would be quite strange not to hypothesize with the reason of growth. I believe that everybody understand that atmospheric field measurements rarely alone are sufficient to answer all the details. Nevertheless, we even say that in the text. "Modelling efforts and

measurement of chemical composition or hygroscopicity of growing mode would be required for unambiguous explanation of the particle growth". Data is / will be available for anyone who have necessary modeling tools, but our message is simply that even if there **can be** mystery compounds or unforeseen mechanisms behind the apparent growth, the explanation **can be** as simple as dilution of emission during transport. We cannot and we do not even try to prove that airmass advection and inhomogeneously distributed sulfuric acid to be 100% certain explanation. But it makes sense, so why to hide that hypothesis. All in all, we think our discussion related to growth is quite appropriate as is now.

**Page 13, line 397:**

**Why is March excluded here? If March does show other origin for NPF compared to the pollution transport, this could be well used to show the difference between anthropogenically and naturally initiated NPF. I do not understand why valuable data are omitted here?**

We excluded March to visualize the true difference between size distributions caused by the airmass origin, not the season. In March, solar radiation is clearly stronger and events more frequent and therefore we think that comparison of these two cases (eastern and western wind cases in $26^{th}$ Jn – $4^{th}$ Feb) to distribution that, on average, represents lighting conditions similar to that of $26^{th}$ Jan – $4^{th}$ Feb, would serve the purpose better. March data would hinder the message. We could also plot the averaged full year spectra, since we naturally have those data also but we think representation is better this way. Alternatively, the whole line (yellow) could be removed since the main message is in the difference between the other two distributions (blue and red)

**Page 13, line 402-403: Give detailed evidence for this statement! What is that based upon?**

MS says now "…our observations on clearly elevated CCN-sized aerosol concentrations in eastern airmasses (Figs. X&X) point toward a clear contribution of Kola Peninsula $SO_2$ emissions to winter-time CCN concentrations in the region."

**Page 14, line 423: Comment: Shorten this sentence and split it in two, it is too long.**

Now: "For better understanding of the contribution of Kola $SO_2$ emissions to local and regional CCN concentrations and upscaling our results to cover the whole (sub)-arctic Eurasia with vastly polluting industrial cities such as Norilsk, require more measurements. Those measurements should be complemented by CCN or cloud residual measurements – ideally in more than only one location (SMEAR I) around the Kola peninsula. Regional chemical transport and aerosol dynamic modeling would be necessary for thorough assessment."

**4 Conclusions**

**Page 14, line 410-418:**

**I am missing some statistics of how often NPF was observed with regard to certain wind directions and concentrations exceeding a threshold of SO2 occurred stating pollution was initiating the process, etc.. Also, such situations should be put into context when conditions were favorable and NPF did not occur. See general comment above!**

Now it reads: "Newly formed (4-10 nm, concentration > 50 cm$^{-3}$) particles were observed in altogether 34 days between $1^{st}$ November 2019 and $15^{th}$ March 2020 out of which ca. 60% were associated with eastern winds or with the period of wind direction change from ~west to east."  For further analysis, we do not have resources right now. This study was not aimed to be a statistical analysis but a mechanistic study. As written in the answer to referee #1 comments, our multidecadal observations are a topic of the next study.

SO2 has no threshold since sulfuric acid makes the particles. Radiation, CS etc. need to be accounted (as in proxy by Dada et al. 2020) as well as temperature that affects the J. Few events reported here is not sufficient to make conclusions on threshold SO2 concentraion. Instead, one should analyse several years / decades of data, classified

based on UVB and maybe CS as well as temperature (and ideally NH3). Then SO2 would possibly have some threshold value. This will be one of the topics of our future study.

**Language comments:**

All below comments have been accounted for and either corrected as suggested or corrected otherwise during proofreading.

**Introduction**

**Page 2, line 46:... precipitates ...**

**Page 2, line 62:... exist ...**

**Page 3, line 66:Leave out ... » of « ...**

**Methods**
**2.1 Site and time of the study**

**Page 3, line 93:**

**The station ...**

**Comment : As a general comment use the article « the » when describing things. I think it is not good grammar saying « Staion is located... ». This shall be « The station is located ... ».**

**Page 4, line 97: ... the closest ...**

**2.2 Instrumentation**

-

**2.3 Nucleation rate calculation**

-

**2.4 Sulfuric acid proxy calculation**

-

**2.5 Trajectory analysis**

-

**3 Results and discussion**

**3.1 New particle formation during the measurement period**

**Page 6, line 170: ... aerosol number size distribution ...**

**Page 6, line 171:... « observed » ... instead of ... « recorded » ...**

**Page 6, line 174:... took place ...**

**Page 6, line 181:... easterly winds ...**

**Page 6, line 182:...westerly winds ...**

**Page 6, line 185:...and were transported ...**

**3.2 Case study 28th – 29th January 2020**

**3.2.1 Meteorological situation and trace gas concentrations**

**3.2.2 Aerosol precursors**

**Page 8, line 230:... are observed ...**

**Page 8, line 231:... remains ...**

**Page 8, line 234:... ends up ...**

**3.2.3 New particle formation**

**Page 8, line 244:... clusters ...**

**Page 10, line 284:... explain ...**

**Page 11, line 341: ... nucleation rates ...**

**Page 12, line 348:... exist ...**

**3.2.4 Particle growth and relevance for CCN-concentrations**

**Page 12, line 353: ... nucleation rates ...**

**Page 12, line 363:... than these ...**

**Page 12, line 375:... in the close environment of emission sources with high .... concentrations ...**

**Page 13, line 377: ... diameters ...**

**Page 13, line 379:... of growing modes ...**

Page 13, line 381:... few tens of ...

Page 13, line 382:... at dìffèrent supersaturations ...

Page 13, line 383:... containing ...

Page 13, line 384:... concentrations ...

Page 13, line 385:.. shows an increase ...

Page 13, line 386: The concentration of particles larger than ...

Page 13, line 387:... these events ...

Page 13, line 390:... a 1-week period of easterly winds ...

Page 13, line 393:... westerly winds ...

Page 13, line 393:... than the average ...

Page 13, line 400:... towards ...

Page 13, line 400:... an accurate ...

Page 13, line 400:... to CCN concentrations ...

Page 13, line 402:... towards ...

**4 Conclusions**

Page 14, line 407:... concentrations of sulfuric acid were high enough ...

Page 14, line 420: ... few nm and larger ...

Page 21, line 655: ... into the atmosphere.  ... processing sites.

Page 22, line 660: ... aerosol number size distribution ... (Comment: include size range)

Page 27, line 693: .... easterly winds ...

Page 27, line 694:  .... westerly winds ...

**Supplement**

Page 1, line 7: ... linear diameter scale.

Page 1, line 10: ... intensive nucleation process ...

 Page 3, line 9: ... particle number size distribution ...

**Page 7, line 10:** ... close to zero ...

**Page 8, line 2:** ... particle number size distribution ...

---

## Author Response (AR2)

Dear authors, dear Mikko,

Thanks for the revised manuscript. One of the reviewers had a another look at the revised manuscript and has some last technical comments (see comments be referee #1).

In addition, I also found some more minor/technical issues. I would encourage one of the native speakers to read the manuscript one more time, there are still some English grammar issues.

**Thank you for the positive comments. Below our responses to the comments by you and referee 1 are marked in bold. Besides addressing these comments, we carefully checked out the grammar of the text once again.**

Here are some examples plus some further comments:

- Title: "sub-arctic" -> "subarctic", **Corrected**

- Line 47-50: Please re-phrase this long and hard-to-understand sentence

**We rewrote the text on these lines into the following form: " Most of the atmospheric sulfate is formed from $SO_2$ in a liquid phase in cloud droplets, and these droplets either evaporate leading to sulfate aerosol production or precipitate as acid rain. However, with very high concentrations of $SO_2$ downwind the Kola Peninsula area, high production rates of gas-phase sulphuric acid ($H_2SO_4$) due to photochemical oxidation of $SO_2$ is expected."**

- Line 100: Add "the" before "Russian". **Corrected**

- Line 110-111: "The other DMPS" ... is this part of the Twin-DMPS or another DMPS? What is the diameter range of the first DMPS?

**Yes, this is part of the Twin-DPMS. We modified the sentence into the following form: "The DMPS measuring 3-10 nm particles malfunctioned during 9-10 and 14-27 January, resulting in the loss of data from this size range on those days."**

- Line 116: Add "of" after "concentrations" and remove the comma. **Corrected**

- Line 120: Maybe better: "THE same calibration coefficient was used for THE reported MSA and HIO3 CONCENTRATIONS". **Corrected, as suggested.**

- Line 149: Is this equation now from Stolzenburg et al, Jokinen et al, or Beck et al?´or a combination of al?

**To clarify this, we re-wrote the text into the following form: "$GR_2$ was approximated by assuming irreversible sulphuric acid condensation as the sole mechanism of growth similar to Jokinen et al. (2018) and Beck et al. (2021), and it was calculated according to the formula given by Stolzenburg et al., (2020):"**

- Line 154: Add "A" before "more" and "the" before "GR". **Added.**

- Line 164: Add "A" or "One" before "drawback" and "the" before "concentration". **Added.**

- Line 165: It should probably be "GFs". **We re-wrote it as "Furthermore, the temporal variability of GR …"**

- Line 169: Probably better: "The effect of the different approaches to determine GF is visualized in ..."

**We agree. Corrected as suggested.**

- Data availability: Please add the DOI/link.

**Done:** https://zenodo.org/record/5524857#.YUyVoGYzbwc

- Figure 3 and 4: The period 01/05 and 03/01 in the first panel should probably be white (now it is interpolated)

**Corrected**

- Figure 6: Panel-labels are missing. I also assume that they can be made smaller/places above each other. What is the reason to show both arrival heights? This is not addressed in the text so far.

**Corrected.**

**Showing 2 arrival heights reveals, for example, how sensitive the calculated trajectories are to the exact arrival height.**

- Figure 7: Units of the colorbar are missing

**Corrected**

- Figure 8: Quite long figure caption and I would suggest to move the discussion within the caption to the main text.

**We removed that last 2 sentences from the figure caption, as roughly the same issues are already discussed in the main text. We think that the rest of the text in this caption is essential for understanding the contents of the figure.**

- Figure 10: Panel-labels are missing.

**Corrected.**

- Supplement: Maybe add the paper title to the top.
- Figure S2: Add a proper figure caption before starting the discussion.

**Done**

- Figure S3, S6, S7, S9: Panel labels are missing.

**Corrected**

- Figure S5 and S8 and S11: Panel labels are missing (in figure and caption).

**Corrected**

Referee #1 comments

I congratulate the authors for the nice job, the quality of the manuscript improved significantly from the first version. The manuscript is now ready for publication after addressing the following minor corrections.

**We thank the referee for the new comments, which further improve the quality of this work. Our responses to each comment are written in bold.**

Line 84: Include also reference to Brean et al 2021 for SA-DMA nucleation (https://doi.org/10.1038/s41561-021-00751-y)

**We added the suggested reference and another recent reference relavant to this statement. The revised text reads now: "Amines, especially dimethyl amine, were found to contribute to the initiation of nucleation in polluted urban air (Yao et al., 2018; Brean et al., 2021; Cai et al., 2021)."**

Line 237: quantify the "several orders of magnitude", how much was the increase?

**Increase is ca. X100. We rewrote "ca. 2 orders of magnitude"**

Line 404: a good reference here would be Kivekäs et al. 2016 https://doi.org/10.3402/tellusb.v68.29706

**We agree. We added this reference.**

Line 479: zenodo link is missing

**Fixed: https://zenodo.org/record/5524857#.YUyVoGYzbwc**

Figure 1: mention in the label that Varrio is the measurement site.

**We added "Our measurement site is located in Värriö." to the figure caption.**

Figure 2: colorscale and subplot labels are missing.

**Corrected.**

Figure 3c and 4c consider using a log scale for sulfuric acid

**We prefer keeping a linear scale to see more clearly the sulphuric acid concentration spikes typically associated with new particle formation events. In a logarithmic scale, low-**

**concentration data points with large uncertainties and not relevant for new particle formation would become the dominant feature of these plots.**

Figure 5 and 7: "examination period" sounds weird, replace with something like "examined NPF events"

**We agree. We rewrote it to the following form: "… during the period 28–29 January 2020."**

Figure 6: subplot labels are missing.

**Corrected**

figure 7f: in the legend you are mixing acronyms (SA) with chemical formula (NH3), consider using a more consistent notation. Additionally, in the caption you should mention that the subscript number after SA indicates the number of SA molecules in each cluster.

**Corrected (SA, IA and MSA substituted by their chemical formulae)**

Figure 8: consider rephrasing the final sentence to mention that the discrepancy is within the expected variability when comparing atmospheric nucleation rate measurements with models based on chamber data.

**We actually removed this sentence from the figure caption, because i) the editors requested moving such text from the figure caption to the main text, and ii) because almost identical text already existed in the main text.**

Figure 9: I would mention that the size of the circles is proportional to the concentration. Replace "an anion" with "the anion".

**Corrected as suggested.**

Figure 11: remove the sentence "New particle formation in the eastern air mass significantly increases the concentrations of particles in every size class." This single sentence is not needed because this figure is discussed in the main text and it is also a bit misleading because it does not mention the role of primary emissions (which is properly discussed in the main text).

**We fully agree on this. We removed the sentence.**